

# Asymptotics of Weil-Petersson volumes
# and two-dimensional quantum gravities

Luca Griguolo[1]⋆, Jacopo Papalini[2]†, Lorenzo Russo[3]‡ and Domenico Seminara[3]°

**1** Dipartimento SMFI, Università di Parma and INFN Gruppo Collegato di Parma,
Viale G.P. Usberti 7/A, 43100 Parma, Italy
**2** Department of Physics and Astronomy, Ghent University,
Krijgslaan, 281-S9, 9000 Gent, Belgium
**3** Dipartimento di Fisica, Università di Firenze and INFN Sezione di Firenze,
via G. Sansone 1, 50019 Sesto Fiorentino, Italy

⋆ luca.griguolo@unipr.it , † jacopo.papalini@ugent.be ,
‡ lorenzo.russo@unifi.it , ° seminara@fi.infn.it

## Abstract

We propose a refined expression for the large genus asymptotics of the Weil-Petersson volumes of the moduli space of super-Riemann surfaces with an arbitrary number of boundaries. Our formula leverages the connection between JT supergravity and its matrix model definition, utilizing some basic tools of resurgence theory. The final result holds for arbitrary boundary lengths and preserves the polynomial structure of the supervolumes. As a byproduct we also obtain a prediction for the large genus asymptotics of generalized Θ-class intersection numbers. We extend our proposal to the case of the quantum volumes relevant for the Virasoro minimal string/Liouville gravity. Performing the classical limit on the quantum volumes, we recover a formula for the ordinary Weil-Petersson building blocks of JT gravity.

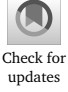

# 1  Introduction

Jackiw-Teitelboim (JT) gravity [1, 2] is a renowned example of two-dimensional quantum gravitational theory, admitting a holographic interpretation [3] and being exactly solvable [4, 5]. In the standard bosonic case, the path integral is constructed from the volume of the moduli of hyperbolic Riemann surfaces, also known as the *Weil-Petersson volume* [6]. When these surfaces have a boundary, the path-integration also involves summing over "wiggles" along boundaries of Riemann surfaces [3, 7, 8]. The wiggles are controlled by the Schwarzian theory [9, 10], constructed with the boundary reparametrization mode. JT gravity has been the subject of an impressive amount of studies, both from the perturbative and the nonperturbative [11–18] point of view, and it has represented a favorite playground to test general ideas on black holes [19, 20], wormholes [21, 22], and holography [23, 24]. More concretely, the theory is defined by the action:

$$\mathcal{S}_{\text{JT}} = -S_0 \chi(\Sigma) - \frac{1}{2}\int_\Sigma \mathrm{d}^2 x\sqrt{g}\,\Phi\,(\mathrm{R}+2) - \int_{\partial\Sigma}\mathrm{d}s\sqrt{h}\,(\mathrm{K}-1)\,, \tag{1}$$

where $\Sigma$ is the two dimensional space-time with boundary $\partial\Sigma$, $\chi(\Sigma)$ is its Euler characteristic, $g$ is the metric and $\Phi$ is a scalar field called dilaton. Path-integrating over the dilaton sets $\mathrm{R}=-2$ requiring the space-time to be hyperbolic and constrains the dynamics of the theory to live on its one-dimensional boundary (see [25] for a nice review). In this set-up it is possible to express the partition function of the theory with $\beta_i$ asymptotic boundaries, $Z(\beta_1,\ldots,\beta_n)$, in terms of Weil-Petersson volumes $V_{g,n}^{\text{WP}}(b_1,\ldots,b_n)$ to every order in the genus expansion. To do this, we first isolate the contribution coming from surfaces with genus $g$ and $n$ Schwarzian boundaries:

$$Z(\beta_1,\ldots,\beta_n)_c = \sum_{g=0}^{+\infty} e^{S_0(2-2g-n)} Z_{g,n}(\beta_1,\ldots,\beta_n)\,. \tag{2}$$

Then we compute $Z_{g,n}(\beta_1,\ldots,\beta_n)$ by glueing $n$ hyperbolic "trumpets" connecting the asymptotic boundaries $\beta_i$ to some geodesic boundaries of length $b_i$, whose contribution is

$$Z^{\text{tr}}(\beta,b) = \frac{e^{-\frac{b^2}{4\beta}}}{\sqrt{4\pi\beta}}\,, \tag{3}$$

to a $n$-boundaries Weil-Petersson volume as explained in [5]:

$$Z_{g,n}(\beta_1,\ldots,\beta_n) = \int_0^{+\infty} b_1\mathrm{d}b_1 Z^{\text{tr}}(\beta_1,b_1)\cdots\int_0^{+\infty} b_n\mathrm{d}b_n Z^{\text{tr}}(\beta_n,b_n)V_{g,n}^{\text{WP}}(b_1,\ldots,b_n)\,. \tag{4}$$

From (2) and (4), it is clear that the form of $V_{g,n}^{\mathrm{WP}}(b_1,\ldots,b_n)$ is of physical importance and that an analysis of the large genus asymptotics of $V_{g,n}^{\mathrm{WP}}(b_1,\ldots,b_n)$ is crucial for understanding the nonperturbative structure of JT gravity. Such an analysis and related ones were conducted in [26–28]. On the other hand, $V_{g,n}^{\mathrm{WP}}(b_1,\ldots,b_n)$ are well known in mathematical literature and have been studied in great detail in the last decades. These volumes are notoriously difficult to compute from scratch, and it is only after Mirzakhani's seminal work [29] that it was possible to systematize their evaluation through some integral recursion relations. Once the genus $g$ and the number of boundaries $n$ are selected, it is indeed possible to obtain the form of the desired volume using Mirzakhani's recursions; however, with few exceptions, a closed-form expression for arbitrary $g$ and $n$ remains elusive. Soon after the discovery of these recursion relations much effort was devoted to the study of $V_{g,n}^{\mathrm{WP}}(b_1,\ldots,b_n)$ in some simplified regime: in particular it was possible to investigate the asymptotic form of $V_{g,n}^{\mathrm{WP}}(b_1,\ldots,b_n)$ as the genus of the involved surfaces increased [30–32]. Along this line, many interesting conjectures were formulated and later proved. To mention a few, in [5], the leading order asymptotic behavior of $V_{g,n}^{\mathrm{WP}}(b)$ was conjectured via a matrix integral approach, in [33] the proposed asymptotics was extended to encompass the case involving an arbitrary number of boundaries via the string equation and in [26] a more refined systematic approach was developed allowing in principle for the inclusion of all the subleading contributions.

JT gravity admits supersymmetric generalizations [11, 34], endowed with different amounts of supersymmetry. In particular, the $\mathcal{N}=1$ case has been thoroughly studied [11], and the path-integral can be performed in terms of a supersymmetric generalization of Weil-Petersson volumes. At the same time, the boundary theory is governed by a super-Schwarzian action [35]. These *super-volumes* are a natural supersymmetric generalization of Weil-Petersson volumes and will be denoted as $V_{g,n}^{\mathrm{SWP}}(b_1,\ldots,b_n)$. They were introduced in [11], where the supersymmetric generalization of JT gravity has been studied and later investigated in mathematics [36]. Similarly to $V_{g,n}^{\mathrm{WP}}(b_1,\ldots,b_n)$, the super-volumes are known to satisfy some integral recursion relations analogous to Mirzakhani's recursions. They are equally difficult to express in a closed form. Moreover, they share with $V_{g,n}^{\mathrm{WP}}(b_1,\ldots,b_n)$ the connection to many different areas of research. In particular, they are closely related to a special class of intersection numbers that generalize the so-called $\Theta$ class, which was introduced in [37]. Finally, as already stated, $V_{g,n}(b_1,\ldots,b_n)$ constitute the building blocks of the computation of the Euclidean path integral of the $\mathcal{N}=1$ supersymmetric generalization of JT gravity.[1]

In this paper, we begin the investigation of the large genus asymptotics of the super-volumes $V_{g,n}^{\mathrm{SWP}}(b_1,\ldots,b_n)$. At present, their large genus asymptotics is only known in the case of one geodesic boundary and was proposed in [11] and refined later in [38]. Much less is known for the super-volumes with $n > 1$ geodesic boundaries. Therefore, we will generalize the conjecture of [11] in this direction. The key highlight of our analysis is preserving the polynomial structure of the Weil-Petersson volumes by our asymptotic formula. Leveraging this result, we can also formulate a prediction for the large genus asymptotics of a generalization of $\Theta$-class intersection numbers.

The paper is organized as follows. In section 2, we present our conjecture for the large genus asymptotics of $V_{g,n}^{\mathrm{SWP}}(b_1,\ldots,b_n)$ concisely, including the first subleading correction to them. Subsequently, in subsection 2.1, we offer some numerical evidence to support it. In subsection 2.2, we use some essential tools of resurgence theory and the nonperturbative equivalence between JT supergravity and a matrix integral to motivate the form of our conjecture. Section 3 is dedicated to the applications. In particular, subsection 3.1 is more mathematically oriented. It is devoted to deriving an asymptotic formula for the generalized $\Theta$-class inter-

---

[1]To be precise we are referring to the theory in which time reversal is not gauged and each spin structure is weighted with $(-1)^\zeta$. With $\zeta$ the Atiyah-Singer index.

section numbers, valid for the large genus. In subsection 3.2, we present another application of our formalism, finding the asymptotics of the quantum volumes $V_{g,n}^{\mathrm{b}}(P_1, \cdots, P_n)$ recently introduced in the study of Liouville gravity/Virasoro minimal string [39]. In subsection 3.3, we take the classical limit of the asymptotics of the quantum volumes to obtain the ordinary $V_{g,n}^{\mathrm{WP}}(b_1, \ldots, b_n)$ relevant for JT gravity. We discuss possible extensions of our investigations in section 4. The paper includes various appendices containing more technical aspects and some subleading computations.

## 2 The large genus asymptotics of $V_{g,n}^{\mathrm{SWP}}(b_1, \ldots, b_n)$

The Weil-Petersson volumes of super Riemann surfaces of genus $g$ with $n$ boundaries $V_{g,n}^{\mathrm{SWP}}(b_1, \ldots, b_n)$ are polynomials in $b_i^2$ of degree $g-1$ for $g > 0$ and play a fundamental role in the path-integral definition of $\mathcal{N} = 1$ super JT gravity [11]. To begin with, let us assume that the Weil-Petersson super-volumes have a well-defined large genus expansion of the following form:

$$V_{g,n}^{\mathrm{SWP}}(b_1, \ldots, b_n) \sim V_{g,n}^{(0)}(b_1, \ldots, b_n) + V_{g,n}^{(1)}(b_1, \ldots, b_n) + \cdots + V_{g,n}^{(i)}(b_1, \ldots, b_n) + \cdots, \qquad (5)$$

where $V_{g,n}^{(0)}(b_1, \ldots, b_n)$ represents the leading order contribution to the asymptotic expansion, while $V_{g,n}^{(i)}(b_1, \ldots, b_n)$ stands for the $i$-th order sub-leading correction. By this, we mean that for genus $g \to \infty$ the following relations hold:

$$\frac{V_{g,n}^{\mathrm{SWP}}(b_1, \ldots, b_n)}{V_{g,n}^{(0)}(b_1, \ldots, b_n)} \sim 1, \qquad \text{and} \qquad \frac{V_{g,n}^{\mathrm{SWP}}(b_1, \ldots, b_n) - \sum_{k=0}^{i-1} V_{g,n}^{(k)}(b_1, \ldots, b_n)}{V_{g,n}^{(i)}(b_1, \ldots, b_n)} \sim 1, \qquad (6)$$

and in addition we expect that:

$$\frac{V_{g,n}^{(i)}(b_1, \ldots, b_n)}{V_{g,n}^{(i-1)}(b_1, \ldots, b_n)} = \mathcal{O}\left(\frac{1}{g}\right). \qquad (7)$$

These relations are not sufficient to uniquely characterize the form of $V_{g,n}^{(i)}(b_1, \ldots, b_n)$ for finite $g$. However, as we will show in the following, there exists a natural functional expression for each $V_{g,n}^{(i)}(b_1, \ldots, b_n)$ motivated by the correspondence between super JT gravity and a matrix model. Interestingly, in Appendix B, the $i$-th term featured in the expansion (5) contributes to the $i$-th loop fluctuation around the one-eigenvalue instanton of the matrix model partition function. In the following, when referring to $V_{g,n}^{(i)}(b_1, \ldots, b_n)$, we will always have in mind such preferred choice. In [11], Witten and Stanford proposed the following formula for leading order asymptotic of $V_{g,1}(b)$:

$$V_{g,1}^{\mathrm{SWP}(0)}(b) \simeq -\frac{\Gamma(2g-1)}{2^g \pi^{3-2g} i} \oint_0 \frac{\mathrm{d}z}{z} \frac{1}{\sin(2\pi z)^{2g-1}} \frac{\sinh(bz)}{b}. \qquad (8)$$

Here, the integral must be performed counterclockwise around the origin. Interestingly, as implied by eq. (8), it yields a polynomial in $b^2$ of degree $g-1$. Furthermore, the coefficient of $b^{2g-2}$ matches, for large values of $g$, with the exact result obtained in [11] for the highest degree term of $V_{g,1}^{\mathrm{SWP}}(b)$. Therefore, it precisely agrees with the leading term of an expansion of

the type described in (5). Building upon this insightful observation, we propose the following generalization in the case of $n$ geodesic boundaries, each with a length denoted as $b_1, \ldots, b_n$, in the regime where $g \gg 1$:

$$V^{(0)}_{g,n}(b_1, \ldots, b_n) \simeq (-1)^n \frac{\pi^{2g+n-4}}{2^{g+1-n}i} \Gamma(2g+n-2) \oint_0 \frac{dz}{z} \frac{1}{\sin(2\pi z)^{2g-2+n}} \prod_i^n \frac{\sinh(b_i z)}{b_i}. \quad (9)$$

The above integral produces a polynomial in $b_i^2$ of degree $g-1$, reducing of course to (8) in the case $n = 1$.

The expression in (9) is suggested by the exact functional equations that relate $V^{\text{SWP}}_{g,n+1}(b_1, \ldots, b_{n+1})$ to $V^{\text{SWP}}_{g,n}(b_1, \ldots, b_n)$ when the length of one boundary is analytically continued to an imaginary value:

$$V^{\text{SWP}}_{g,n+1}(b_1, \ldots, b_n, 2\pi i) = -(2g-2+n)V^{\text{SWP}}_{g,n}(b_1, \ldots, b_n). \quad (10)$$

These recursion relations were proven in [36] using the representation (38) of the super-volumes and the pull-back properties of some cohomology classes, and they bear resemblance to the ones used in [33] to derive the asymptotics of Weil-Petersson volumes with $n$ boundaries. Our formula can be obtained by starting from the natural ansatz:

$$V^{(0)}_{g,n}(b_1, \ldots, b_n) = C_{g,n} \oint_0 \frac{dz}{z} \frac{1}{\sin(2\pi z)^{2g-2+n}} \prod_i^n \frac{\sinh(b_i z)}{b_i}, \quad (11)$$

and using (10) to derive a recursion relation for $C_{g,n}$. Assuming the initial value $C_{g,1}$ as provided by (8), we easily arrive at:

$$C_{g,n} = (-1)^n \frac{\pi^{2g+n-4}}{2^{g+1-n}i} \Gamma(2g+n-2).$$

The subtle part of the derivation lies in justifying the ansatz (11), for which we will offer a non-rigorous argument in its favor in sec. 2.2, based on the formulation of JT supergravity in terms of a matrix integral. Additionally, we propose the following form for the first sub-leading term in (5):

$$V^{(1)}_{g,n}(b_1, \ldots, b_n) = (-1)^{n-1} \frac{\pi^{2g+n-5}}{2^{g+3-n}i} \Gamma(2g-3+n) \oint \frac{dz}{z^2} \frac{1}{\sin(2\pi z)^{2g-3+n}} \prod_{i=1}^n \frac{\sinh(b_i z)}{b_i}. \quad (12)$$

In analogy to (9), this expression results in a polynomial of degree $g-1$ in $b_i^2$, and its explicit form is formally derived in Appendix B through an examination of the contributions from one-eigenvalue instantons to the n-point correlators in SJT gravity. We present a systematic method to possibly obtain analogous formulas for all the other sub-leading corrections to $V^{(0)}_{g,n}(b_1, \ldots, b_n)$.

We expect our formulas (9) and (12) to be reliable for $g \gg 1$ and arbitrary boundary lengths. In particular, since these integrals produce polynomials in $b_i^2$, we expect these polynomials to contain every monomial present in the true expression of $V^{\text{SWP}}_{g,n}(b_1, \ldots, b_n)$. Furthermore, each coefficient of the monomials computed using our formulas should be a good approximation to the corresponding coefficient of $V^{\text{SWP}}_{g,n}(b_1, \ldots, b_n)$. Below, we provide a closed-form expression for these coefficients, and in the next section, we numerically verify the claim above.

The detailed analysis of (9) and (12) is presented in Appendix A. There, given the expansion of $V^{\text{SWP}}_{g,n}(b_1, \ldots, b_n)$ in monomials $b_1^{2\alpha_1} \ldots b_n^{2\alpha_n}$, expressed as:

$$V^{\text{SWP}}_{g,n}(b_1, \ldots, b_n) = \sum_{\substack{\alpha_1, \ldots, \alpha_n \geqslant 0 \\ \sum_i \alpha_i \leqslant g-1}} c_{\alpha_1, \ldots, \alpha_n}(g) b_1^{2\alpha_1} \cdots b_n^{2\alpha_n}, \quad (13)$$

we derive the following asymptotic prediction for the coefficients of these polynomials:

$$c_{\alpha_1,\dots,\alpha_n}(g) \sim \overset{(0)}{c}_{\alpha_1,\dots,\alpha_n}(g) + \overset{(1)}{c}_{\alpha_1,\dots,\alpha_n}(g) + \dots, \tag{14}$$

with

$$\overset{(0)}{c}_{\alpha_1,\dots,\alpha_n}(g) = \frac{(2g-3+n)!}{(-1)^{n+g-|\alpha|-1}} \frac{2^{g-2-4|\alpha|}\pi^{2g-3-2|\alpha|}}{(2g-2-2|\alpha|)!} \mathcal{B}^{2g-2+n}_{2g-2-2|\alpha|}\left(\frac{2g-2+n}{2}\right) \prod_{i=1}^{n} \frac{1}{(2\alpha_i+1)!}, \tag{15a}$$

$$\overset{(1)}{c}_{\alpha_1,\dots,\alpha_n}(g) = \frac{(2g-4+n)!}{(-1)^{n+g-|\alpha|}} \frac{2^{g-3-4|\alpha|}\pi^{2g-4-2|\alpha|}}{(2g-2-2|\alpha|)!} \mathcal{B}^{2g-3+n}_{2g-2-2|\alpha|}\left(\frac{2g-3+n}{2}\right) \prod_{i=1}^{n} \frac{1}{(2\alpha_i+1)!}. \tag{15b}$$

Here $\mathcal{B}^a_k(x)$ are the generalized Bernoulli polynomials of degree $k$ [40] and $|\alpha| = \sum_{i=1}^{n} \alpha_i$. As a consistency check for our results, by using the asymptotic behaviours:

$$\begin{aligned}
\mathcal{B}^{2g-2+n}_{2g-2-2|\alpha|}\left(\frac{2g-2+n}{2}\right) &\sim -(2g-2|\alpha|-2)! \frac{2^{-2g-n+3}\pi^{2\alpha+n-\frac{3}{2}}\cos(\pi(|\alpha|+g+n))}{(-1)^n\sqrt{g-|\alpha|-1}}, \\
\mathcal{B}^{2g-3+n}_{2g-2-2|\alpha|}\left(\frac{2g-3+n}{2}\right) &\sim -(2g-2|\alpha|-2)! \frac{2^{-2g-n+4}\pi^{2\alpha+n-\frac{5}{2}}\cos(\pi(|\alpha|+g+n))}{(-1)^n\sqrt{g-|\alpha|-1}},
\end{aligned} \tag{16}$$

valid when $g \gg 1$, it is easy to verify that the relation (7) is satisfied as expected.

We conclude this section by providing a simple formula for (9) in the regime $g \gg b_i$. In this case, the integral is dominated by the saddle points located at $z = \pm\frac{1}{4}$ and (9) evaluates to:

$$V^{(0)}_{g,n}(b_1,\dots,b_n) \sim (-1)^n \frac{\Gamma\left(2g+n-\frac{5}{2}\right)}{2^{g-\frac{3}{2}-n}\pi^{\frac{9}{2}-n-2g}} \prod_i^n \frac{\sinh\left(\frac{b_i}{4}\right)}{b_i}. \tag{17}$$

We notice that the structure of this result is quite similar to the one obtained by [33] in the analysis of the large genus asymptotics of $V^{\text{WP}}_{g,n}(b_1,\dots,b_n)$, the main difference being that (17) features $\sinh(b_i/4)$ instead of $\sinh(b_i/2)$.

## 2.1 Numerical evidences

We now report some numerical evidence that supports our conjecture for the asymptotic behavior of $V^{\text{SWP}}_{g,n}(b_1,\dots,b_n)$. For this purpose, it is convenient to define with $c^{\text{asymp}}_{\alpha_1,\dots,\alpha_n}(i|g)$ the truncation of the asymptotic expansion (14) including up to the $i$-th term. We compare the expression of the predicted coefficients $c^{\text{asymp}}_{\alpha_1,\dots,\alpha_n}(i|g)$ of the super-volumes obtained from (15a) with the actual coefficients $c_{\alpha_1,\dots,\alpha_n}(g)$ of $V^{\text{SWP}}_{g,n}(b_1,\dots,b_n)$. We calculated these coefficients by adapting to our case the Mathematica notebook attached to [39], where the Mirzakhani's recursions were efficiently implemented to compute the quantum volumes, a generalization of Weil-Petersson volumes $V^{\text{WP}}_{g,n}(b_1,\dots,b_n)$ introduced in [39].

Let us define $\mathcal{R}_{\alpha_1,\dots,\alpha_n}(i|g)$ to be the ratio between $c_{\alpha_1,\dots,\alpha_n}(g)$ and $c^{\text{asymp}}_{\alpha_1,\dots,\alpha_n}(i|g)$. We denote with $\mathcal{R}^{(n)}(i|g)$ the ratio corresponding to the worst approximation to the true value $c_{\alpha_1,\dots,\alpha_n}(g)$, meaning:

$$\left|\mathcal{R}^{(n)}(i|g) - 1\right| = \max_{\alpha_1,\dots,\alpha_n} \left|\mathcal{R}_{\alpha_1,\dots,\alpha_n}(i|g) - 1\right|. \tag{18}$$

In Figure 1, we present a plot depicting the values of $\mathcal{R}^{(2)}(i|g)$ for both $i = 0$ and $i = 1$ across a range of genera spanning from 1 to 23. The green dots on the graph correspond to the results for $i = 0$, while the blue dots represent the case of $i = 1$.

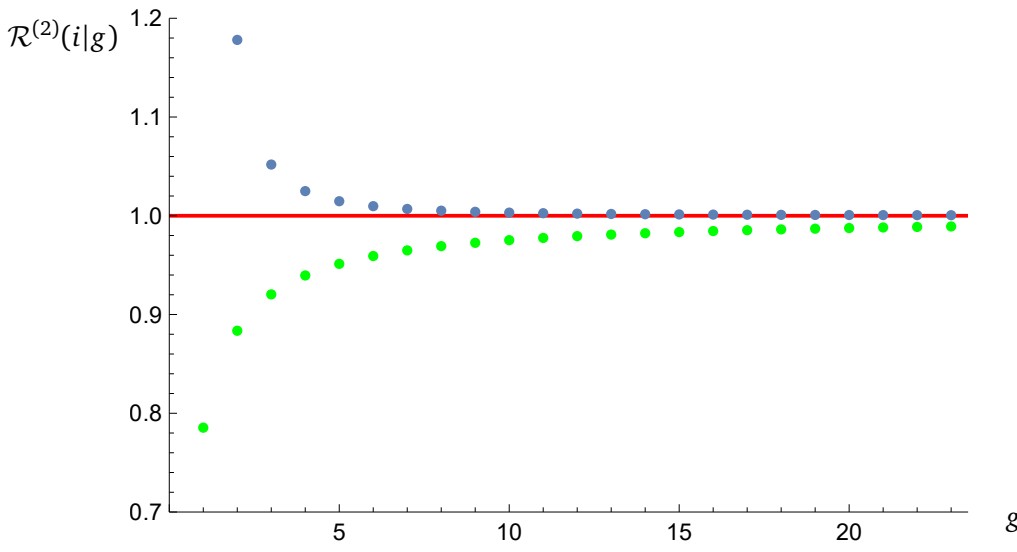

Figure 1: $\mathcal{R}^{(2)}(i|g)$ for $i = 0, 1$ as the genus varies.

Our results demonstrate that the coefficients predicted by our formulas (9) and (12) offer an excellent approximation to the actual coefficients of $V_{g,n}^{\text{SWP}}(b_1, b_2)$, even for small genera. To illustrate this, let us first focus on the case when $i = 0$. When $g = 1$, the value of $V_{1,2}^{\text{SWP}}(b_1, b_2)$ is constant, independently of the values of $b_1$ and $b_2$. The discrepancy between our estimate $c_{0,0}^{(0)}(1)$ and the exact value $c_{0,0}(1)$ is slightly above 20%. However, as the genus increases, our approximation consistently and rapidly improves. For instance, when $g = 23$, the error decreases to around 0.7%. When we include the subleading correction by setting $i = 1$, our approximation becomes even more accurate and does so more rapidly. We observe an error of approximately 1% at $g = 6$, which further decreases to just 0.05% when $g = 23$.

Alternatively, in Figure 2, we have plotted the ratio between the polynomial in $b$ corresponding to the super-volume $V_{g,2}^{\text{SWP}}(b, b)$ and our asymptotic expression (9) (with $b_1 = b_2 = b$) for genera $g = 5, 10$, and 20. The results are represented by solid lines in gray, blue, and red, respectively, as we vary the values of $b$ from 1 to 20. Within the same figure, we have also depicted the ratio between $V_{g,2}^{\text{SWP}}(b, b)$ and the asymptotic expression (17), which is valid for $g \gg b_i$ (again with $b_1 = b_2 = b$). These results are represented by dashed lines, using the same color correspondence of the previous case for the different genera.

As the genus increases, both (9) and (17) become closer to the true value of the super-volumes with two boundaries, suggesting that they are good asymptotic formulas for $V_{g,n}^{\text{SWP}}(b_1, b_2)$. However, the approximation (9) consistently yields superior results compared to (17) when varying the boundary lengths. This outcome is as expected because (9) not only provides an estimate for $V_{g,n}^{\text{SWP}}(b_1, \ldots, b_n)$ as a generic function but also faithfully replicates its polynomial structure, producing accurate approximations for each of its coefficients. Importantly, this accuracy is shown to be independent of the lengths of the boundaries. Lastly, we remark that in the analysis above, we have only presented the results for the case $n = 2$ to simplify the discussion; however, similar behaviors hold for any value of $n$.

## 2.2 The asymptotic formula made easy

In this section, we try to motivate the form of the ansatz (11) by simple arguments, without any claim of rigor. To better understand our reasoning, we will need to introduce some basic concepts of resurgence theory (we defer to [41] for a more comprehensive discussion) and the duality between JT supergravity and a matrix integral [11].

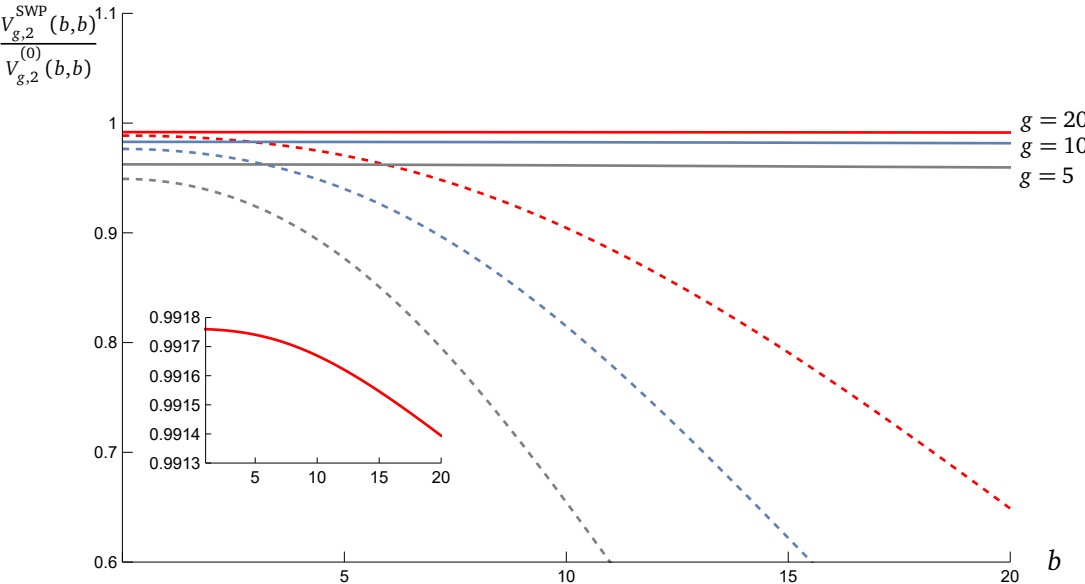

Figure 2: The gray, blue and red solid line represents respectively the ratio $V^{\text{SWP}}_{g,2}(b,b)/V^{(0)}_{g,2}(b,b)$ for genera genera $g = 5, 10$ and 20. The dashed curve describes the ratio where we have replaced $V^{(0)}_{g,2}(b,b)$ with its approximation (17) valid for $g \gg b_i$. We also display a zoomed-in version of the red line that shows that the accuracy of our approximation is remarkably constant as the lengths of the boundaries are varied.

Roughly speaking, the aim of resurgence theory is to make sense of formal power series with a vanishing radius of convergence. This kind of series is ubiquitous in physics and arises very often when we perform a perturbative expansion in some parameter of the theory. For this reason, it is necessary to develop a machinery that systematically produces well defined functions from these formal objects. Resurgence theory addresses this problem in two steps: firstly, it defines an auxiliary series called Borel transform $\mathcal{B}$ and then deduces a well-defined expression for the original formal series through the analysis of the singularities of $\mathcal{B}$ in the complex Borel plane.

To be more specific, let us consider an asymptotic power series that we assume to be resurgent, meaning that the form of its coefficients allows to conduct the program of resurgence theory:

$$\phi(u) = \sum_{k=0}^{+\infty} a_k u^k \,, \tag{19}$$

with $a_k \sim k!$ so that the series has a vanishing radius of convergence. In the context of resurgence theory, we can assign a well-defined meaning to this formal expression through the analysis of the following auxiliary series

$$\mathcal{B}[\phi](t) = \sum_{k=1}^{+\infty} \frac{a_k}{\Gamma(k)} t^{k-1} \,, \tag{20}$$

which is called Borel transform of $\phi$ and has a finite radius of convergence. In particular, we define the completion of the original asymptotic series to be the Borel resummation of $\phi$,

meaning:

$$\mathcal{S}_\theta[\phi](u) = a_0 + \int_{\mathcal{C}_\theta} \mathrm{d}t \, e^{-t/u} \mathcal{B}[\phi](t). \tag{21}$$

Here, the integral is carried out in the complex plane of $t$ along a radial path defined by an angle $\theta$ relative to the positive real semi-axis. This angle is chosen so that the path avoids all potential singularities. It is straightforward to verify that (21) reproduces the original formal series (19) when we integrate (20) term by term.

In this context, the nonperturbative information complementing (19) is intricately tied to the ambiguity stemming from our choice of the integration contour. This ambiguity becomes apparent whenever the path we select crosses a singularity of $\mathcal{B}[\phi](t)$, and we must decide how to navigate around it. Consequently, we can isolate a particular nonperturbative sector present in (21) by opting for a contour encompassing a specific Borel transform singularity.

It is worth noting that the above argument works both ways. Suppose that the explicit form of the Borel resummation for $\phi(u)$ can be obtained from a certain nonperturbative correction, the contribution of which can be expressed as a contour integral of the type (21). In such a case, one can extract the perturbative information embedded in the coefficients of the asymptotic series for $\phi(u)$ by applying Cauchy's theorem to $\mathcal{B}[\phi](t)$:

$$a_k = \frac{\Gamma(k)}{2\pi i} \oint_0 \mathrm{d}t \, \frac{\mathcal{B}[\phi](t)}{t^k}. \tag{22}$$

This statement looks rather trivial, but it will be the key ingredient to deriving the ansatz (11) for the large genus asymptotics of $V_{g,n}^{\mathrm{SWP}}(b_1, \ldots, b_n)$.

The strategy is to identify an observable within a physical theory, whose perturbative series expansion incorporates the super-volumes. By doing so, if we are fortunate enough to derive an explicit expression for the Borel transform of such a series, we can retrieve the form of $V_{g,n}^{\mathrm{SWP}}(b_1, \ldots, b_n)$ through the relation (22). This program, while quite ambitious, is achievable, albeit in an approximate form.

The physical theory under investigation is JT supergravity, and the observable requiring analysis is the Euclidean path integral of the theory. Within this context, the super-volumes $V_{g,n}^{\mathrm{SWP}}(b_1, \ldots, b_n)$ serve a similar role to that of $V_{g,n}^{\mathrm{WP}}(b_1, \ldots, b_n)$ in JT gravity. Indeed, the connected part of $Z(\beta_1, \ldots, \beta_n)$ is still perturbatively described by a genus expansion as presented in (2), with the perturbative parameter of the theory being $e^{-S_0}$. Furthermore, the contribution arising from a surface with genus $g$ and $n$ asymptotic boundaries to this observable closely resembles (4); it is obtained by joining $n$ hyperbolic super-trumpets with a form that is

$$Z_{\mathrm{SJT}}^{\mathrm{tr}}(\beta, b) = \frac{1}{\sqrt{2\pi\beta}} e^{-\frac{b^2}{4\beta}}, \tag{23}$$

to a $n$-boundaries Weil-Petersson super-volume as shown in [11]:

$$Z_{g,n}^{\mathrm{SJT}}(\beta_1, \ldots, \beta_n) = \int_0^{+\infty} b_1 \mathrm{d}b_1 \, Z_{\mathrm{SJT}}^{\mathrm{tr}}(\beta_1, b_1) \cdots \int_0^{+\infty} b_n \mathrm{d}b_n \, Z_{\mathrm{SJT}}^{\mathrm{tr}}(\beta_n, b_n) V_{g,n}(b_1, \ldots, b_n). \tag{24}$$

From this perspective, deducing a closed-form expression for the Borel transform of the genus expansion is an extremely challenging task. This is the reason why we must turn to an alternative formulation of the theory, one that utilizes matrix integrals. In this framework, the nonperturbative aspects become more accessible. Below, for completeness, we will introduce specific elements of the matrix model description of JT supergravity.

To fix the notation, given a function $\mathcal{O}(M)$ of a random variable that takes values in an ensemble of $N \times N$ matrices, we define the expected value of $\mathcal{O}$ as

$$\langle \mathcal{O} \rangle \equiv \int \mathrm{d}M \, \mathcal{O}(M) e^{-N \mathrm{Tr} V(M)}. \tag{25}$$

Here $V(M)$ is the potential of the matrix integral, and $dM$ is the appropriate measure that depends on the choice of the ensemble of matrices. For our purpose, we need to consider the so-called Altland-Zirnbauer ensemble $(1, 2)$ and, in this case (25) simplifies to an integral over the $N$ eigenvalues of the matrices (see [42] for more details). Remarkably it was demonstrated in [11] that there is a perturbative equivalence between such a matrix integral, with a suitable choice of the potential, and JT supergravity.[2] In detail, the correspondence between the two theories has the following dictionary:

$$Z(\beta_1, \ldots, \beta_n) \longleftrightarrow 2^n \langle \operatorname{Tr} e^{-\beta_1 M} \cdots \operatorname{Tr} e^{-\beta_n M} \rangle, \tag{26}$$

where the prefactor is required because of the different normalization of the spectral densities of the theories. This reformulation of JT supergravity is very helpful to our objective since, in a matrix integral, it is possible to access the nonperturbative physics systematically [43]. In particular, by leveraging this fact, it is possible to extract the leading nonperturbative correction to the genus expansion of the path integral, which is controlled by configurations in the matrix integral where one eigenvalue tunnels away from the region where it is perturbatively confined [44]. Then, by removing the super-trumpets, we can access the first instantonic contribution $\mathcal{V}_n^{[1]}(b_1, \ldots, b_n; S_0)$ to the generating function of the super-volumes $\mathcal{V}_n^{[0]}(b_1, \ldots, b_n; S_0)$ which is formally defined by:

$$\mathcal{V}_n^{[0]}(b_1, \ldots, b_n; S_0) = \sum_{g=0}^{+\infty} e^{S_0(2-2g-n)} V_{g,n}^{\text{SWP}}(b_1, \ldots, b_n). \tag{27}$$

A similar analysis was conducted in the context of JT gravity in [5, 26] and in the context of the Virasoro minimal string in [39].

In a generic one-cut matrix model these leading nonperturbative contributions are controlled by configurations in which one eigenvalue sits in the classically forbidden region and all the others are in the allowed one. It is then natural to expect that these contributions have an integral representation of the following form:

$$\int_{\mathcal{I}} dE \langle \rho(E) \rangle \ldots, \tag{28}$$

where $\langle \rho(E) \rangle$ is the spectral density of the matrix model in the forbidden region and is determined by analysing the influence of the other eigenvalues on the one that tunneled in the forbidden region and "$\ldots$" stands for the appropriate insertion depending on the observable under examination. Finally, $\mathcal{I}$ represents a contour in the forbidden region. Following this logic and the analysis in [5, 26, 39], it is easy to argue that the leading order nonperturbative contribution to $\mathcal{V}_n^{[0]}(b_1, \ldots, b_n; S_0)$, that we denote with $\mathcal{V}_n^{[1]}(b_1, \ldots, b_n; S_0)$, has the following integral representation:

$$\mathcal{V}_n^{[1]}(b_1, \ldots, b_n; S_0) \propto \int_{\mathcal{I}} dE \langle \rho(E) \rangle \prod_{i=1}^{n} \frac{\sinh(b_i \sqrt{-E})}{b_i} \propto \int_{\mathcal{I}} \frac{dE}{E} e^{-V_{\text{eff}}(E)} \prod_{i=1}^{n} \frac{\sinh(b_i \sqrt{-E})}{b_i}, \tag{29}$$

where $\mathcal{I}$ is a contour that goes to infinity in the complex plane, passing through the closest stationary point of $V_{\text{eff}}(E)$ to the origin,[3] while $V_{\text{eff}}(E)$ is the effective potential that one eigenvalue of the matrix integral feels. The effective potential $V_{\text{eff}}(E)$ is the combination of the original

---

[2]Actually this is not completely true, the equivalence requires a double scaling limit from the matrix integral side, but we will not need to be precise here.

[3]There is some arbitrariness in the choice of the integration contour which is the reflection of the nonperturbative ambiguity of the present description. Nonetheless, we will not address this aspect since our use of (29) will be mainly formal.

potential $V(E)$ and the repulsive contribution coming from the matrix integral measure, and it can be shown to have the following form:

$$V_{\text{eff}}(E) = 2e^{S_0} \int_0^{-E} dx \frac{\cos\left(2\pi\sqrt{-x}\right)}{\sqrt{-2x}} = -\frac{\sqrt{2}\sin\left(2\pi\sqrt{-E}\right)}{\pi} e^{S_0}. \tag{30}$$

The proportionality symbol in (29) means that we are suppressing some numerical prefactors.

The idea is now to reinterpret the integral (29) as the contribution to the Borel resummation of $\mathcal{V}_n^{[0]}(b_1,\ldots,b_n;S_0)$ encoding information from the initial nonperturbative sector. We aim to reformulate the integral (29) into a contour integral of the form (21). In this reformulation, the path is expected to encompass the primary relevant singularity of the Borel transform. To do so, we start by expressing (29) using the new variable $z = \sqrt{-E}$ :

$$\mathcal{V}_n^{[1]}(b_1,\ldots,b_n;S_0) \propto \int_{\tilde{\mathcal{I}}} \frac{dz}{z} e^{-V_{\text{eff}}(z)} \prod_{i=1}^n \frac{\sinh(b_i z)}{b_i}, \tag{31}$$

then by letting $t = e^{-S_0} V_{\text{eff}}(z)$ we arrive at the expression:

$$\mathcal{V}_n^{[1]}(b_1,\ldots,b_n;S_0) \propto \int_{\hat{\mathcal{I}}} dt\, e^{-e^{S_0}t(z)} \frac{1}{z(t)V'_{\text{eff}}(z(t))} \prod_{i=1}^n \frac{\sinh(b_i z(t))}{b_i}, \tag{32}$$

from which we can immediately identify the Borel transform of $\mathcal{V}_n^{[0]}(b_1,\ldots,b_n;S_0)$ to be

$$\mathcal{B}[\mathcal{V}_n^{[0]}](t) \propto \frac{1}{z(t)V'_{\text{eff}}(z(t))} \prod_{i=1}^n \frac{\sinh(b_i z(t))}{b_i}. \tag{33}$$

As previously stressed, our derivation remains essentially formal for several reasons. Firstly, we have exclusively relied on the leading contribution to the complete nonperturbative sector of $\mathcal{V}_n^{[0]}(b_1,\ldots,b_n;S_0)$. Consequently, we can only regard (33) as an approximation of the true Borel transform. Furthermore, we intentionally omitted the examination of the new integration contour resulting from the change in the integration variable. Lastly, it is important to note that the relation $t = e^{-S_0} V_{\text{eff}}$ is invertible only within the interval $[-1/2, 1/2]$. Clearly, additional caution is required to justify our interpretation.

Nonetheless, if we insist in identifying (33) as the Borel transform of the generating function $\mathcal{V}_n^{[0]}(b_1,\ldots,b_n;S_0)$, we can obtain the ansatz (11) for $V_{g,n}^{\text{SWP}}(b_1,\ldots,b_n)$ by using the relation in (22) and switching to the $z$ variable:

$$V_{g,n}^{\text{SWP}}(b_1,\ldots,b_n) \propto \oint_0 dt \frac{\mathcal{B}[\mathcal{V}_n^{[0]}](t)}{t^{2g-2+n}} \propto \oint_0 \frac{dz}{z} \frac{1}{\sin(2\pi z)^{2g-2+n}} \prod_i^n \frac{\sinh(b_i z)}{b_i}. \tag{34}$$

For those concerned about the perceived lack of systematicity in the final part of this section, we will address this issue in Appendix B. There, we refine our analysis and introduce a more systematic, although still formal, approach for computing instantonic contributions in a matrix integral [15]. Importantly, by following this method, we successfully replicate the result presented in (9) with all the accurate prefactors. Furthermore, we outline the procedure for predicting all subleading corrections to it. Specifically, we apply this methodology to the first subleading correction, yielding the result already presented in (12).

## 3 Applications and related results

We are now ready to extend the results of the previous section to three strictly related topics. Firstly, we formulate a prediction for the large genus asymptotics of a generalization of $\Theta$-class

intersection numbers, the mathematical objects underlying the structure of the super-volumes. Then, we apply a method, essentially analogous to the one employed in subsection 2.2, to obtain a polynomial asymptotic formula for the quantum volumes $V_{g,n}^{\mathrm{b}}(P_1, \ldots, P_n)$, introduced in [39] in the context of Liouville gravity/Virasoro minimal string. Finally, from this last result, we deduce a similar formula for the ordinary Weil-Petersson volumes $V_{g,n}^{\mathrm{WP}}$.

### 3.1 Asymptotic behaviour of generalized $\Theta$-class intersection numbers

For clarity and completeness, we introduce a few basic concepts of algebraic geometry. Let $\Sigma_{g,n}$ be a closed Riemann surface of genus $g$ and with $n$ marked points $p_1, \ldots, p_n$ and let $\mathcal{M}_{g,n}$ be the moduli space of $\Sigma_{g,n}$. Further, let $\overline{\mathcal{M}}_{g,n}$ be the Deligne-Mumford compactification of $\mathcal{M}_{g,n}$. We can define $n$ line-bundles $\mathcal{L}_i$ over $\mathcal{M}_{g,n}$ (one for each marked point) that extend over $\overline{\mathcal{M}}_{g,n}$ and whose fiber is the cotangent space to $\Sigma_{g,n}$ at $p_i$. In this context the usual Weil-Petersson volumes $V_{g,n}^{\mathrm{WP}}(b_1, \ldots, b_n)$ are computed in terms of $\psi$-class intersection numbers through the following relation [29]:

$$V_{g,n}^{\mathrm{WP}}(b_1, \ldots, b_n) = \int_{\overline{\mathcal{M}}_{g,n}} \exp\left(2\pi^2 \kappa\right) \exp\left(\frac{1}{2}\sum_{i=1}^{n} b_i^2 \psi_i\right), \tag{35}$$

where $\kappa$ is the first Miller-Morita-Mumford class which is related to the Weil-Petersson symplectic form by $\omega = 2\pi^2 \kappa$. Moreover, $\psi_i = c_1(\mathcal{L}_i)$ is the first Chern class of the complex line bundle $\mathcal{L}_i$. Interestingly, (35) implies that $V_{g,n}^{\mathrm{WP}}(b_1, \ldots, b_n)$ is a polynomial in $b_i^2$ of degree $3g + n - 3$ and provides a bridge between Weil-Petersson volumes and intersection theory. As a matter of fact, exploiting the above equality, the large genus asymptotics of the intersection numbers

$$\langle \tau_{d_1} \cdots \tau_{d_n} \rangle_g = \int_{\overline{\mathcal{M}}_{g,n}} \psi_1^{d_1} \cdots \psi_n^{d_n}, \tag{36}$$

whose explicit expression for generic $g$ and $n$ is in principle very hard to obtain, was conjectured in [45] and recently proved in [46].

In the following, we shall use (9) and (12) to formulate a conjecture about the large genus asymptotics of a different kind of intersection numbers, recently considered by [36]. They are obtained from the product of the $\psi$-class, the first Miller-Morita-Mumford class $\kappa$ and a different cohomology class called $\Theta$-class:

$$\langle \tau_{d_1}, \ldots, \tau_{d_n}; m \rangle_\kappa^\Theta = \int_{\overline{\mathcal{M}}_{g,n}} \Theta_{g,n} k^m \prod_{i=1}^{n} \psi^{d_i}. \tag{37}$$

The first two classes where defined in the discussion above, while for the geometrical meaning of $\Theta_{g,n}$, we defer to [47]. For our purpose, it is sufficient to know that $\Theta_{g,n}$ is a cohomology class of real dimension $4g - 4 + 2n$, so that in (37), we have to impose the constraint $m + |d| = g - 1$ [with $|d| = \sum_{i=1}^{n} d_i$].

To obtain a conjecture for the asymptotics of (37), we exploit a relation that connects the three cohomology classes above with the super-volumes [36], which is analogous of (35) but in the supersymmetric context:

$$V_{g,n}^{\mathrm{SWP}}(b_1, \ldots, b_n) = (-1)^n \int_{\overline{\mathcal{M}}_{g,n}} \Theta_{g,n} \exp\left(\pi^2 \kappa\right) \exp\left(\frac{1}{4}\sum_{i=1}^{n} b_i^2 \psi_i\right). \tag{38}$$

The right-hand side of the equality evaluates to:

$$V_{g,n}^{\text{SWP}}(b_1,\ldots,b_n)=\sum_{\substack{\alpha_1,\ldots,\alpha_n\geqslant 0 \\ \sum_i \alpha_i \leqslant g-1}}\frac{(-1)^n 4^{-|\alpha|}\pi^{2g-2-2|\alpha|}}{(g-1-|\alpha|)!\prod_{i=1}^{n}\alpha_i!}\langle\tau_{\alpha_1},\ldots,\tau_{\alpha_n};g-1-|\alpha|\rangle_\kappa^\Theta\, b_1^{2\alpha_1}\cdots b_n^{2\alpha_n}. \tag{39}$$

Therefore, by comparing the expression (39) with (13), we obtain the following prediction for the large genus asymptotics of the intersection numbers:

$$\langle\tau_{\alpha_1},\ldots,\tau_{\alpha_n};g-1-|\alpha|\rangle_\kappa^\Theta \sim (-1)^n\frac{(g-1-|\alpha|)!\prod_{i=1}^{n}\alpha_i!}{4^{-|\alpha|}\pi^{2g-2-2|\alpha|}}c_{\alpha_1,\ldots,\alpha_n}(g). \tag{40}$$

If we use the explicit form (15a) of the leading term $c_{\alpha_1,\ldots,\alpha_n}^{(0)}(g)$ in the expansion of $c_{\alpha_1,\ldots,\alpha_n}(g)$, we find immediately the dominant contribution for $\langle\tau_{\alpha_1},\ldots,\tau_{\alpha_n};g-1-|\alpha|\rangle_\kappa^\Theta$ in the large genus expansion:

$$\sim \frac{\Gamma(2g-2+n)}{(-1)^{g-|\alpha|-1}}\mathcal{B}_{2g-2-2|\alpha|}^{2g-2+n}\left(\frac{2g-2+n}{2}\right)\frac{2^{g-2-2|\alpha|}}{\pi}\frac{(g-1-|\alpha|)!}{(2g-2-2|\alpha|)!}\prod_{i=1}^{n}\frac{\alpha_i!}{(2\alpha_i+1)!}. \tag{41}$$

Exploiting the well-known identity $n!/(2n)!=2^{-n}/(2n-1)!!=2^{-2n}\sqrt{\pi}/\Gamma\left(n+\frac{1}{2}\right)$, we can rewrite the above expression as follows

$$\sim -\frac{1}{(-2)^{g+|\alpha|}\sqrt{\pi}}\frac{\Gamma(2g-2+n)}{\Gamma\left(g-|\alpha|-\frac{1}{2}\right)}\mathcal{B}_{2g-2-2|\alpha|}^{2g-2+n}\left(\frac{2g-2+n}{2}\right)\prod_{i=1}^{n}\frac{1}{(2\alpha_i+1)!!}. \tag{42}$$

The first sub-leading correction is obtained when we instead insert $c_{\alpha_1,\ldots,\alpha_n}^{(1)}(g)$ given by (15b) into (40). After some trivial algebra, we get

$$\sim -\frac{1}{(-2)^{g+|\alpha|+1}\sqrt{\pi}}\frac{\Gamma(2g-3+n)}{\Gamma\left(g-|\alpha|-\frac{1}{2}\right)}\mathcal{B}_{2g-2-2|\alpha|}^{2g-3+n}\left(\frac{2g-3+n}{2}\right)\prod_{i=1}^{n}\frac{1}{(2\alpha_i+1)!!}. \tag{43}$$

A more refined analysis about the asymptotics of $\langle\tau_{\alpha_1},\ldots,\tau_{\alpha_n};0\rangle_\kappa^\Theta$ was recently conducted on a more rigorous ground in [46], where in particular it was shown that:

$$\langle\tau_{\alpha_1},\ldots,\tau_{\alpha_n};0\rangle_\kappa^\Theta \sim \frac{\Gamma(2g-2+n)}{\pi\, 2^{2g-1}}\left(1-\frac{1}{2(2g-3+n)}+\ldots\right)\prod_{i=1}^{n}\frac{1}{(2\alpha_i+1)!!}. \tag{44}$$

If we take into account that $\mathcal{B}_0^a(x)=1$, the sum of our predictions (42) and (43) exactly reduces to (44) in the particular case $|\alpha|=g-1$.

## 3.2 The asymptotic form of the quantum volumes $V_{g,n}^{\text{b}}(P_1,\ldots,P_n)$

The methodology used in the realm of SJT gravity can be employed to investigate another significant two-dimensional gravity theory, namely Liouville gravity [39, 48–51]. The latter is a non-critical bosonic string theory in two dimensions, defined by the sum of three conformal field theories on the worldsheet, namely a spacelike Liouville CFT with central charge $c=1+6(b+b^{-1})^2\geqslant 25$, a timelike Liouville CFT with central charge $\hat{c}\leqslant 1$ playing the role of the matter and the usual $bc$-ghost system accounting for gauge fixing with central charge $c_{\text{ghost}}=-26$. As usual, the central charge $\hat{c}$ is constrained to be $\hat{c}=26-c$ to cancel the conformal anomaly. This theory has been extensively studied on the disk topology, characterizing its connection to quantum groups and its possible rewording as a 2d dilaton gravity with

sinh-dilaton potential [48, 49]. Both formulations correspond to precise deformations of the familiar SL(2, ℝ) JT gravity. The power of quantum groups also enables to determine the form of a dynamical action realizing the quantum deformation of the Schwarzian boundary description, the so-called q-Schwarzian [52], which plays the role of the boundary dual of Liouville gravity [53].

In [39], there has been instead a remarkable attempt for correctly interpreting Liouville gravity on general Riemann surfaces. In this context, one would like to make sense of string scattering amplitudes corresponding to worldsheet CFT correlators integrated over moduli space $\mathcal{M}_{g,n}$ of genus-$g$ Riemann surfaces with $n$ punctures, i.e.[4]

$$V_{g,n}^{\mathrm{b}}(P_1,\ldots,P_n) \equiv \int_{\mathcal{M}_{g,n}} Z_{\mathrm{gh}} \left\langle V_{P_1}\cdots V_{P_n}\right\rangle_g \left\langle \hat{V}_{iP_1}\cdots \hat{V}_{iP_n}\right\rangle_g, \qquad (45)$$

where $P_k$ is the Liouville momentum associated to the vertex operators $V_{P_k}$[5] of the spacelike Liouville theory, while the mass-shell condition implies $\hat{P}_k = iP_k$ for the corresponding vertex operators $\hat{V}_{iP_k}$ of the timelike Liouville.

Notably, it was pointed out in [39] that, despite the complicated form of the string amplitudes in (45), the quantities $V_{g,n}^{\mathrm{b}}(P_1,\cdots,P_n)$ admit a much simpler representation in terms of the loop equations of a double-scaled matrix model with leading density of states given by $\rho_0^{(b)}(E)\mathrm{d}E = 2\sqrt{2}\frac{\sinh(2\pi b\sqrt{E})\sinh(2\pi b^{-1}\sqrt{E})}{\sqrt{E}}\,\mathrm{d}E$. The correspondence of the latter with the universal Cardy density of primaries in a two-dimensional CFT led then the authors to rename the theory as Virasoro minimal string and call the amplitudes (45) as the quantum volumes, which indeed correspond to the quantum generalization of the standard Weil-Petersson volumes in JT gravity. The loop equations can therefore be translated into a deformed version of the Mirzakhani recursion relation obeyed by the $V_{g,n}^{\mathrm{b}}(P_1,\ldots,P_n)$, through which the string worldsheet amplitudes are perturbatively fully determined by the double-scaled matrix integral description.

However, as it happens for the matrix model of JT gravity [5], there are nonperturbative completions to the quantum volumes generating function that, in parallel with the JT story, can be identified with one-eigenvalue instantons. We can, therefore, exploit the general techniques exposed in Appendix B to extract these one-instanton corrections corresponding to one eigenvalue sitting at a nonperturbative saddle of the effective matrix model potential. In formulae:

$$\mathcal{V}_n^{(1)}(z_1,\ldots,z_n)\Big|_{g_s^0} = \frac{1}{4\pi}\int_\gamma \frac{\mathrm{d}z}{z}\left(\prod_{k=1}^n \frac{\sqrt{2}\sin(4\pi P_k z)}{P_k}\right) e^{-\frac{2\sqrt{2}}{g_s}\left(\frac{\sin(2\pi \hat{Q}z)}{\hat{Q}} - \frac{\sin(2\pi Q z)}{Q}\right)}, \qquad (46)$$

with $Q = b^{-1}+b$, $\hat{Q}=b^{-1}-b$ and again $g_s = e^{-S_0}$. We observe once again the appearance in the exponent of the effective potential $S_0(z)$, obtained by applying (B.5) to this case. The study of the nonperturbative effects encoded in (46) brought the authors of [5] to state the following large-genus asymptotics of the quantum volumes [$g \gg 1$ and $g \gg P_k$ for any $k$]:

$$V_{g,n}^{\mathrm{b}}(P_1,\ldots,P_n) \simeq \frac{\prod_{j=1}^n \frac{\sqrt{2}\sinh(2\pi b P_j)}{P_j}}{2^{\frac{3}{2}}\pi^{\frac{5}{2}}(1-b^4)^{\frac{1}{2}}} \times \left(\frac{4\sqrt{2}b\sin(\pi b^2)}{1-b^4}\right)^{2-2g-n}\Gamma\left(2g+n-\frac{5}{2}\right). \qquad (47)$$

Following the same steps that lead to the formal derivation of the asymptotic SWP volumes in Section 2.2, we instead arrive at the following formula for the asymptotic value of the quantum

---

[4]$Z_{\mathrm{gh}}$ is the correlator of the $bc$-ghost system.

[5]It corresponds to a primary operator of conformal weight $h_{P_k} = \frac{Q^2}{4} + P_k^2$ with $Q = b^{-1}+b$. We take $b \in [0,1]$.

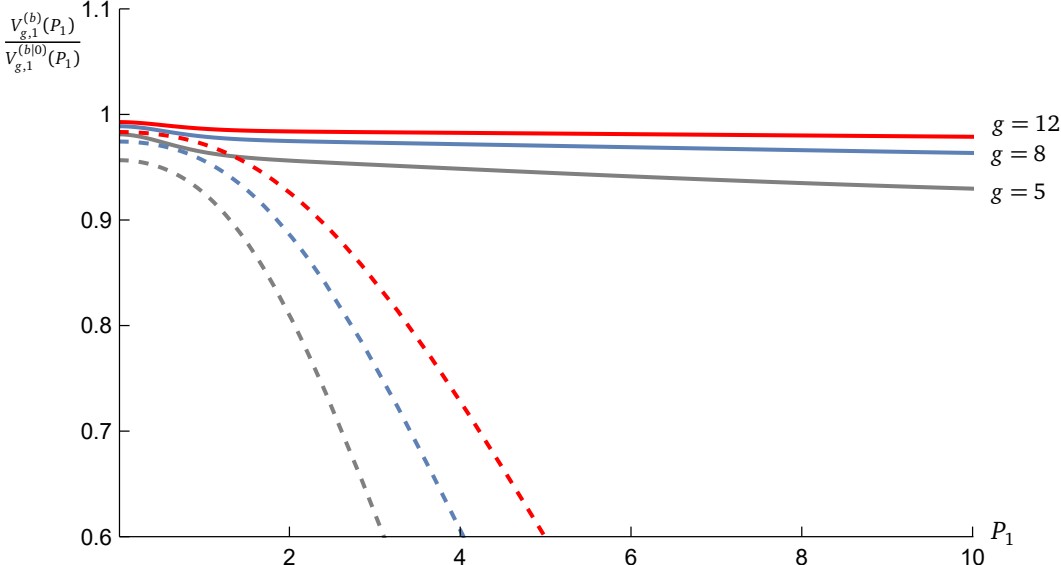

Figure 3: $V_{g,1}^{(b|0)}(P_1)$ represents the asymptotic approximation (48) for normal lines, while to (47) for dashed lines. Red, blue and gray lines correspond respectively to genus $g = 5, 8, 12$. The parameter $b$ was set to $b = \frac{1}{2}$. We notice our asymptotic formula (48) is better and stable for any value of $P_1$.

volumes:

$$V_{g,n}^{(b|0)}(P_1,\ldots,P_n) = \frac{\Gamma(2g+n-2)}{4\pi} \oint \frac{dz}{2\pi i} \frac{1}{z\left(2\sqrt{2}\left(\frac{\sin(2\pi \hat{Q}z)}{\hat{Q}} - \frac{\sin(2\pi Qz)}{Q}\right)\right)^{2g-2+n}}$$
$$\times \prod_{k=1}^{n} \frac{\sqrt{2}\sinh(4\pi P_k z)}{P_k}. \tag{48}$$

The superscript $(b|0)$ indicates that this quantity represents the leading-order asymptotic behavior. However, next-to-leading-order corrections could, in principle, be extracted by considering higher-loop corrections to (46), which correspond to $\mathcal{O}(g_s)$ fluctuations around the instanton background. These corrections could be derived using the methods outlined in Appendices B and B.1.

We immediately point out that in the regime where $g \gg P_i$, the integral (48) can be evaluated using the steepest descent approximation, leading to the result (47). However, the contour integral prescription in (48) actually allows us to recover a true polynomial form of the quantum volumes, with the correct expected degree. In fact, through similar calculations to the ones performed in Appendix A, the integral can be evaluated via a residue and leads to the following closed analytic expression:

$$V_{g,n}^{(b|0)}(P_1,\ldots,P_n) = \sum_{\substack{\alpha_1,\ldots,\alpha_n \geqslant 0 \\ \sum_i \alpha_i \leqslant 3g-3+n}} c_{\alpha_1,\cdots,\alpha_n}^{(b|0)}(g)\, P_1^{2\alpha_1} P_2^{2\alpha_2} \cdots P_n^{2\alpha_n}, \tag{49}$$

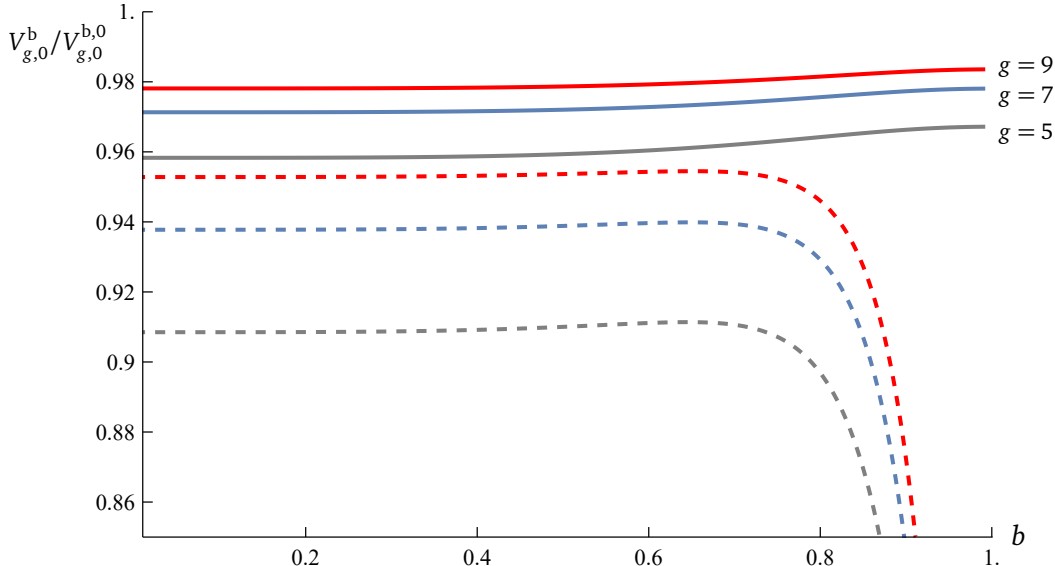

Figure 4: $V_{g,0}^{(b|0)}$ represents the asymptotic approximation (48) for normal lines, while to (47) for dashed lines. Red, blue, and gray lines correspond respectively to genus $g = 5, 7, 9$. We can notice our approximation is better and stable for any value of $b$.

with the coefficients $c_{\alpha_1,\dots,\alpha_n}^{(b|0)}$ explicitly given by $\left(|\alpha| \equiv \sum_{i=1}^{n} \alpha_i \text{ as usual}\right)$

$$c_{\alpha_1,\dots,\alpha_n}^{(b|0)}(g) = A_{g,n,|\alpha|} \prod_{i=1}^{n} \frac{1}{(2\alpha_i + 1)!} \tag{50}$$
$$\times \sum_{k=0}^{6(g-1)+2(n-|\alpha|)} \Gamma(k + 2g - 2 + n) B_{6(g-1)+2(n-|\alpha|),k}\left(x_1, \dots, x_{6(g-1)+2(n-|\alpha|)-k+1}\right),$$

where $B_{n,k}(x_1, \dots, x_{n-k+1})$ are the incomplete Bell polynomials with

$$x_s = \frac{3((-1)^s + 1)}{4} \frac{s!}{(s+3)!} \left(\hat{Q}^{s+2} - Q^{s+2}\right).$$

Moreover, the overall factor is $A_{g,n,|\alpha|} = (-1)^{g+n+1-|\alpha|} \frac{3^{2g+n-2} 2^{2|\alpha|-5g-n+3}}{\pi(6(g-1)+2(n-|\alpha|))!}$. We have also checked the consistency of our results numerically. As it is shown in Figure 3 and Figure 4, our analytic predictions (50) for $V_{g,0}^{(b|0)}$ and $V_{g,1}^{(b|0)}(P_1)$ based on (48) agree very well with the exact quantum volumes, even at low genus. Compared with (47), our results show much better stability as a function of the Liouville momenta and central charge.

## 3.3 The JT limit and the result for $V_{g,n}^{\text{WP}}(\ell_1, \dots, \ell_n)$

As a final application, we observe that the exact expression (50) can be used to deduce the corresponding classical Weil-Petersson volumes $V_{g,n}^{\text{WP}}$ relevant for JT gravity, by performing the limit

$$V_{g,n}^{\text{WP}}(\ell_1, \dots, \ell_n) = \lim_{b \to 0} (8\pi^2 b^2)^{3g+n-3} V_{g,n}^{\text{b}}(P_1, \dots, P_n),$$

where $\ell_i$ are the geodesic lengths related to the Liouville momenta through $P_i = \frac{\ell_i}{4\pi b}$ [39]. If we write the general structure of the Weil-Petersson polynomial as

$$V_{g,n}^{\text{WP}}(\ell_1, \ldots, \ell_n) = \sum_{\substack{\alpha_1,\ldots,\alpha_n \geqslant 0 \\ |\alpha| \leqslant 3g-3+n}} c_{\alpha_1,\cdots,\alpha_n}^{(\text{WP})}(g)\, \ell_1^{2\alpha_1} \ell_2^{2\alpha_2} \cdots \ell_n^{2\alpha_n},$$

we obtain the following result:

$$c_{\alpha_1,\ldots,\alpha_n}^{(\text{WP}|0)}(g) = \frac{\left(8\pi^2\right)^{3g-3+n}}{(4\pi)^{2|\alpha|}} A_{g,n,|\alpha|} \prod_{i=1}^{n} \frac{1}{(2\alpha_i + 1)!} \tag{51}$$

$$\times \sum_{k=0}^{6(g-1)+2(n-|\alpha|)} \Gamma(k+2g-2+n) B_{6(g-1)+2(n-|\alpha|),k}\left(\tilde{x}_1, \ldots, \tilde{x}_{6(g-1)+2(n-|\alpha|)-k+1}\right),$$

The above formula can be inferred directly from (50) by exploiting some properties of the Bell polynomials and we have defined

$$\tilde{x}_s = -\frac{3\left((-1)^s + 1\right)}{2} \frac{1}{(s+1)(s+3)}\,.$$

## 4 Concluding remarks

In this paper, we studied the large genus asymptotics of the Weil-Petersson volume of the moduli space of super Riemann surfaces $V_{g,n}^{\text{SWP}}(b_1, \ldots, b_n)$. By leveraging the duality between JT supergravity and a matrix integral, we formulated an ansatz for the asymptotic form of $V_{g,n}^{\text{SWP}}(b_1, \ldots, b_n)$ and then, by utilizing some functional relations between the super-volumes, we proposed a precise conjecture. The predicted super-volumes showed an excellent numerical agreement to the actual super-volumes $V_{g,n}^{\text{SWP}}(b_1, \ldots, b_n)$ even for low genera, and we demonstrated this fact both by comparing each coefficient of the two polynomials and by analyzing their behavior as their boundary lengths were varied. By exploiting the connection between $V_{g,n}^{\text{SWP}}(b_1, \ldots, b_n)$ and a generalization of $\Theta$-class intersection numbers, we formulated a conjecture for the large genus asymptotics of the latter object. The asymptotics of a simplified version of these intersection numbers was recently studied in [46], and we verified that our prediction reduces to theirs at the leading order. We have also applied our conjecture to the case of Virasoro string/Liouville gravity refining the asymptotic estimates for the quantum volumes defined in [39]. Also in this case we found a fine agreement with their exact evaluation, being stable when varying the Liouville momenta. As emphasized previously, our derivation of the asymptotic formulas relies on a formal treatment of the pertinent integration contours. Nonetheless, the consistent efficacy of our conjecture across various scenarios implies a deeper rationale behind our assumptions. A rigorous mathematical proof of our results is beyond the scope of this paper but it would be certainly welcome. As far as concerned with physics, there are a couple of other playground in which to try our computation strategy: recently [54] a supersymmetric version of Virasoro string has been studied, showing a certain analogy with SJT gravity. A matrix model formulation has been also presented there and some nonperturbative properties have been explored. It would be nice to understand the asymptotic behavior of the related super quantum volumes, as well as to establish their relation with an appropriate intersection theory. The other case of study could be the $\mathcal{N} = 2$ SJT gravity [34], in which the behaviour of the appropriate super-volumes has not been fully scrutinized yet. Another perspective is related to the stringy formulation of Virasoro string: in this case the quantum volumes describe worldsheet amplitudes with the string momenta playing the role

of the geodesic boundary lengths. Our asymptotics preserves the polynomial character in the external momenta and therefore could be useful to understand better truly stringy perturbation theory at large order. Finally, we emphasize that understanding the large-order behavior of Weil-Petersson volumes could prove crucial for diagnosing the late-time behavior of correlators measured in black hole backgrounds within low-dimensional gravities. Specifically, it may offer insights into the spectral form factor, serving as a useful toy model for the thermal two-point function [55–57].

## Acknowledgments

**Funding information** This work has been supported in part by the Italian Ministero dell'Università e della Ricerca (MIUR), and by Istituto Nazionale di Fisica Nucleare (INFN) through the "Gauge and String Theory" (GAST) research project. JP acknowledges financial support from the European Research Council (grant BHHQG-101040024). Funded by the European Union. Views and opinions expressed are however those of the author(s) only and do not necessarily reflect those of the European Union or the European Research Council. Neither the European Union nor the granting authority can be held responsible for them.

## A  Dissecting the asymptotic structure of $V_{g,n}^{\text{SWP}}(b_1, \ldots, b_n)$

In this Appendix, we analyze in detail the polynomial structure of $V_{g,n}^{\text{SWP}}(b_1, \ldots, b_n)$ as predicted from the asymptotic formulas (9) and (12) that we report here for convenience:

$$V_{g,n}^{(0)}(b_1, \ldots, b_n) = (-1)^n \frac{\pi^{2g+n-4}}{2^{g+1-n}i} \Gamma(2g+n-2) \oint_0 \frac{dz}{z} \frac{1}{\sin(2\pi z)^{2g-2+n}} \prod_i^n \frac{\sinh(b_i z)}{b_i}, \quad \text{(A.1)}$$

$$V_{g,n}^{(1)}(b_1, \ldots, b_n) = (-1)^{n-1} \frac{\pi^{2g+n-5}}{2^{g+3-n}i} \Gamma(2g-3+n) \oint \frac{dz}{z^2} \frac{1}{\sin(2\pi z)^{2g-3+n}} \prod_{i=1}^n \frac{\sinh(b_i z)}{b_i}. \quad \text{(A.2)}$$

In the following, we will explicitly determine the form of $V_{g,n}^{(0)}(b_1, \ldots, b_n)$. However, the analysis can be easily generalized for the case of $V_{g,n}^{(1)}(b_1, \ldots, b_n)$.

The integral around the origin in (A.1) reduces, of course, to a residue, therefore we can write:

$$V_{g,n}^{(0)}(b_1, \ldots, b_n) = \frac{(-1)^n \pi^{2g+n-3}}{2^{g-n}} \Gamma(2g+n-2) \text{Res}_{z=0} \left( \frac{1}{z \sin(2\pi z)^{2g-2+n}} \prod_i^n \frac{\sinh(b_i z)}{b_i} \right). \quad \text{(A.3)}$$

To obtain an explicit expression for the polynomial in (A.3) it is necessary to express the product of functions inside the brackets as a Laurent series around $z = 0$ and then select the coefficient of $z^{-1}$. The only non-obvious Laurent series among these functions pertains to the reciprocal of the sine function and reads

$$\left( \frac{1}{\sin(2\pi z)} \right)^{2g-2+n} = \left( \frac{1}{2\pi z} \right)^{2g-2+n} \sum_{k=0}^{+\infty} \mathcal{B}_k^{2g-2+n} \left( \frac{2g-2+n}{2} \right) \frac{(4\pi i z)^k}{k!}, \quad \text{(A.4)}$$

where $\mathcal{B}_k^a(x)$ are the generalized Bernoulli polynomials of degree $k$, whose generating function is:

$$\left( \frac{z}{e^z - 1} \right)^a e^{xz} = \sum_{k=0}^{+\infty} \mathcal{B}_k^a(x) \frac{z^k}{k!}. \quad \text{(A.5)}$$

These polynomials and their properties are very well known in the mathematical literature, and in particular, they satisfy [58]:

$$\mathcal{B}_k^a(x) = (-1)^k \mathcal{B}_k^a(a-x) \,, \tag{A.6}$$

which in turn ensures that (A.4) is a power series with real coefficients as it should be. It is now sufficient to recall the standard Taylor series of $\sinh(bz)$ and the form of the Cauchy product of $n$ series, which is a discrete convolution that allows to merge the product of $n$ power series into a single one. In detail, given $n$ infinite series with complex terms $\{a_{i,k_i}x^{k_i}\}_{i\in\mathbb{N}}$, their product can be expressed as single series in the following way:

$$\prod_{i=1}^{n}\sum_{k_i\geqslant 0} a_{i,k_i}x^{k_i} = \sum_{k=0}^{+\infty} c_k x^k, \quad \text{where} \quad c_k = \sum_{\substack{k_1,\dots,k_n\geqslant 0 \\ k_1+k_2+\cdots+k_n=k}} a_{1,k_1}a_{2,k_2}\dots a_{n,k_n}. \tag{A.7}$$

Using these identities, it is now possible to evaluate the residue in (A.3) explicitly and to analyze the predicted structure of $V_{g,n}^{\text{SWP}}(b_1,\dots,b_n)$.

As a warm up, we begin by considering the case of a single boundary. Setting $n=1$ in (A.3) we arrive at the following expression:

$$V_{g,1}^{(0)}(b) = -\frac{\Gamma(2g-1)}{2^{3g-2}\pi}\sum_{k=g-1}^{2g-2}\mathcal{B}_{2k-2g+2}^{2g-1}\left(\frac{2g-1}{2}\right)\frac{(4\pi i)^{2k-2g+2}}{(2k-2g+2)!}\frac{b^{4g-2k-4}}{(4g-2k-3)!}\,. \tag{A.8}$$

By inspection, it is clear that (A.8) is an even polynomial in $b^2$ of degree $g-1$ and that all of its coefficients are non-vanishing. This precisely matches the true structure of $V_{g,1}^{\text{SWP}}(b)$ as demonstrated in [36]. From this expression, it is now immediate to extract the coefficient of a generic monomial of the form $b^{2\alpha}$ that we denote with $c_\alpha^{(0)}(g)$:

$$c_\alpha^{(0)}(g) = (-1)^{g-\alpha}\frac{\Gamma(2g-1)}{(2\alpha+1)!}\mathcal{B}_{2g-2-2\alpha}^{2g-1}\left(\frac{2g-1}{2}\right)\frac{2^{g-2-4\alpha}\pi^{2g-3-2\alpha}}{(2g-2-2\alpha)!}\,. \tag{A.9}$$

In the more general case of $n$ boundaries, it is still possible to obtain a closed-form expression from (A.3), and we display it below:

$$V_{g,n}^{(0)}(b_1,\dots,b_n) = \frac{(-1)^n\Gamma(2g+n-2)}{2^{3g-2}\pi}$$
$$\times \sum_{k=g-1}^{2g-2}\mathcal{B}_{2k-2g+2}^{2g-2+n}\left(\frac{2g-2+n}{2}\right)\frac{(4\pi i)^{2k-2g+2}}{(2k-2g+2)!}\sum_{\substack{s_1,\dots,s_n\geqslant 0 \\ \sum_i s_i=2g-2-k}}\frac{b_i^{2s_i}}{(2s_i+1)!}\cdots\frac{b_n^{2s_n}}{(2s_n+1)!}\,. \tag{A.10}$$

It is clear that (A.10) is a manifestly symmetric polynomial in $b_i^2$ of degree $g-1$ with non-vanishing coefficients, therefore it reproduces the expected structure of the Weil-Petersson super-volumes. In the present case the coefficient of a generic monomial of the form $b_1^{2\alpha_1}\cdots b_n^{2\alpha_n}$ is denoted with $c_{\alpha_1,\dots,\alpha_n}^{(0)}(g)$ and can be immediately read from above:

$$c_{\alpha_1,\dots,\alpha_n}^{(0)}(g) = \frac{\Gamma(2g-2+n)}{(-1)^{n+g-|\alpha|-1}}\mathcal{B}_{2g-2-2|\alpha|}^{2g-2+n}\left(\frac{2g-2+n}{2}\right)\frac{2^{g-2-4|\alpha|}\pi^{2g-3-2|\alpha|}}{(2g-2-2|\alpha|)!}\prod_{i=1}^{n}\frac{1}{(2\alpha_i+1)!}\,, \tag{A.11}$$

where we used the shorthand notation $|\alpha|=\alpha_1+\cdots+\alpha_n$ along with the condition $0\leq|\alpha|\leq g-1$.

In an analogous fashion we can obtain a closed-form expression for the first sub-leading contribution in (A.2):

$$
\begin{aligned}
V_{g,n}^{(1)}(b_1,\ldots,b_n) &= (-1)^{n-1}\frac{\Gamma(2g+n-3)}{\pi 2^{3g-1}} \\
&\times \sum_{k=g-1}^{2g-2} \mathcal{B}_{2k-2g+2}^{2g-3+n}\left(\frac{2g-3+n}{2}\right)\frac{(4\pi i)^{2k-2g+2}}{(2k-2g+2)!}\sum_{\substack{s_1,\ldots,s_n\geqslant 0 \\ \sum_i s_i=2g-2-k}}\frac{b_i^{2s_i}}{(2s_i+1)!}\cdots\frac{b_n^{2s_n}}{(2s_n+1)!}.
\end{aligned}
\tag{A.12}
$$

Again we can see that the polynomial structure of the above expression correctly reproduces the one expected for a super-volume, and we can extract the expression of the coefficient of a generic monomial appearing in (A.12) that we denote with $c_{\alpha_1,\ldots,\alpha_n}^{(1)}(g)$:

$$
c_{\alpha_1,\ldots,\alpha_n}^{(1)}(g) = \frac{\Gamma(2g-3+n)}{(-1)^{n+g-|\alpha|}}\mathcal{B}_{2g-2-2|\alpha|}^{2g-3+n}\left(\frac{2g-3+n}{2}\right)\frac{2^{g-3-4|\alpha|}\pi^{2g-4-2|\alpha|}}{(2g-2-2|\alpha|)!}\prod_{i=1}^{n}\frac{1}{(2\alpha_i+1)!}.
\tag{A.13}
$$

## B  Systematic approach to one-eigenvalue instantons in JT supergravity

In [26] a general method was developed in order to extract the one-instanton contribution to n-point correlators in JT gravity, including loop corrections around it. Below, we will exploit the same type of formalism, extended to the case of SJT gravity to formally derive the asymptotic formulas (9) and (12) presented in the main text. To this task, we first review some of the basic notations. Let us start by considering the one-instanton correction to the free energy [26]:

$$
\mathcal{F}^{(1)} = \frac{1}{2\pi}\int_\gamma dx\,\psi(x),
\tag{B.1}
$$

where $\gamma$ is the steepest descent contour passing through the non-trivial nonperturbative maximum $x^*$ of the effective potential[6] and $\psi(x)$ is the wavefunction corresponding to having the remaining $N-1$ eigenvalues sitting at the perturbative minimum of the potential,[7] obtained through

$$
\psi(x) = \int_{\gamma_0}\prod_{i=1}^{N-1}d\lambda_i\Delta^2(\lambda_1,\ldots,\lambda_{N-1},x)e^{-\frac{1}{g_s}\sum_{i=1}^{N-1}V(\lambda_i)}e^{-\frac{V(x)}{g_s}}.
\tag{B.2}
$$

Here $\gamma_0$ represents the perturbative contour and $g_s \equiv e^{-S_0}$. As briefly stated below (27), equation (B.1) immediately follows from the definition of the free energy in the matrix model once that we allow one eigenvalue to sit outside of the perturbative cut.

Following [26] the wave function can be parametrized in the following way:

$$
\psi(x(z)) = \exp\left(S(z,g_s)\right),
\tag{B.3}
$$

where $x(z) = z^2$ and the function $S(z,g_s)$ can be organized as a sum weighted by the Euler characteristic $\chi$, i.e.

$$
S(z,g_s) = \sum_{\chi=0}^{+\infty}S_\chi(z)g_s^{\chi-1}.
\tag{B.4}
$$

---

[6]Actually we will not need to work concretely with this contour since we will perform formal steps where we neglect the information about $\gamma$.

[7]This formula is the same as (28) of the main text but written using a different notation: here $\psi(x)$ is what in the main text was called $\langle\rho(E)\rangle$.

The objects $S_\chi(z)$ are in turn defined as:

$$S_\chi(z) = \sum_{\substack{g \geqslant 0, n \geqslant 1 \\ 2g-2+n=\chi-1}} \frac{F_{g,n}(z)}{n!}, \qquad F_{g,n}(z) = \int_{-z}^{z} \cdots \int_{-z}^{z} \omega_{g,n}, \tag{B.5}$$

where $\omega_{g,n}(z_1, \cdots, z_n) = \hat{W}_{g,n}(z_1, \cdots, z_n) \, dz_1 \cdots dz_n$ are the differential forms associated to the resolvents $\hat{W}_{g,n}(z_1, \cdots, z_n)$. Crucially, the resolvents are related to the Weil-Petersson volumes through simple multi-Laplace transforms as

$$\hat{W}_{g,n}(z_1, \ldots, z_n) = 2^{\frac{n}{2}} \prod_{i=1}^{n} \int b_i \, db_i \, e^{-z_i b_i} V_{g,n}^{\text{SWP}}(b_1, \ldots, b_n). \tag{B.6}$$

Actually, we are interested in computing the one-instanton contribution to the n-point correlator, which can be obtained by acting with the so-called loop operator $\Delta_{z_i}$ on the one-instanton sector of the free energy [26]. Defining rigorously the loop operator is beyond the scope of the present paper and we defer to [26] for a more detailed explanation. What's crucial to the present work is that the action of such an operator increases the number of boundaries of the correlators. Specifically, we will have $n$-insertions, one for each boundary, acting on the free energy instanton (B.1), i.e.

$$\hat{W}_n^{(1)}(z_1, \ldots, z_n) = \left( \prod_{k=1}^{n} g_s \Delta_{z_k} \right) \mathcal{F}^{(1)} = \frac{1}{\pi} \int_\gamma dz \, z \left( \prod_{k=1}^{n} g_s \Delta_{z_k} \right) e^{S(z)}, \tag{B.7}$$

where the action of the loop operator is expressed by

$$\Delta_z \omega_{g,n}(z_1, \ldots, z_n) = \omega_{g,n+1}(z_1, \ldots, z_n, z), \tag{B.8}$$

and $\Delta_z$ acts as the usual derivative, obeying the Leibniz rule.

Since we are interested in including the first loop correction to the one-instanton sector, we will need to compute $S(z)$ up to order $g_s$ using the definitions (B.4) and (B.5) applied to SJT gravity. The basic results that we need in our calculation are the following: first, from the relevant spectral density

$$y(z) = -\sqrt{2} \frac{\cos(2\pi z)}{z}, \tag{B.9}$$

we immediately recover the expression $\omega_{0,1}(z_1) = -\frac{1}{2} y(z_1) dx(z_1) = \sqrt{2} \cos(2\pi z_1) dz_1$. The form of $\omega_{0,2}$ is instead universal

$$\omega_{0,2}(z_1, z_2) = \frac{dz_1 dz_2}{(z_1 - z_2)^2}, \tag{B.10}$$

while a peculiar behaviour of SJT gravity is encoded into

$$\omega_{0,n}(z_1, \ldots, z_n) = 0, \tag{B.11}$$

for any $n \geqslant 3$ [11]. In order to compute $S_2(z)$, we also need the explicit result

$$\omega_{1,1}(z_1) = -\frac{\sqrt{2}}{8z_1^2} dz_1. \tag{B.12}$$

Assumed these formulae, some simple computations lead to:

$$S(z) = \frac{\sqrt{2}}{\pi g_s} \sin(2\pi z) - \log(4z^2) + g_s \frac{\sqrt{2}}{4z} + \ldots \tag{B.13}$$

Let us now act with the loop operators on $\psi(x(z^2))$ as in (B.7). It is convenient first to work with a single loop operator on the wave function, i.e.

$$\hat{W}_n^{(1)}(z_1,\ldots,z_n) = \frac{1}{\pi} \int_\gamma \mathrm{d}z \; z \left( \prod_{k=1}^{n-1} g_s \Delta_{z_k} \right) \left( g_s \Delta_{z_n} S(z) \right) e^{S(z)}. \tag{B.14}$$

Then, applying the rule (B.8), we can evaluate the action of $\Delta_{z_n}$ up to order $g_s^2$:

$$\begin{aligned} g_s \Delta_{z_n} S(z) &= \Delta_{z_n} F_{0,1}(z) + \frac{g_s}{2} \Delta_{z_n} F_{0,2}(z) + g_s^2 \Delta_{z_n} \left( F_{1,1}(z) + \frac{1}{6} F_{0,3}(z) \right) + \ldots \\ &= \frac{2z}{z_n^2 - z^2} - g_s^2 \frac{1}{2z z_n^2} + \mathcal{O}\left(g_s^3\right), \end{aligned} \tag{B.15}$$

where we used that $\Delta_{z_n} F_{0,2}(z) = \Delta_{z_n} F_{0,3}(z) = 0$ and $\omega_{1,2}(z_1, z_2) = \frac{1}{4 z_1^2 z_2^2} \mathrm{d}z_1 \mathrm{d}z_2$. The remaining $n-1$ loop operators in (B.14) can act on the exponential $\exp(S(z))$ or on $\Delta_{z_n} S(z)$. However one can easily show that, for instance,

$$g_s^2 \Delta_{z_{n-1}} \Delta_{z_n} = \mathcal{O}(g_s^3), \tag{B.16}$$

since $\Delta_{z_{n-1}} \Delta_{z_n} F_{0,1}(z) = 0$. Therefore, any of these terms where $j$ loop operators act on $\Delta_{z_n} S(z)$ is suppressed by a power of $g_s^{2+j}$. We conclude that the leading contribution, of order $g_s^0$, to the one-instanton sector of the n-point resolvent comes from having all loop insertions operating on $e^{S(z)}$, keeping only the first term in (B.15) for each of them:

$$\hat{W}_n^{(1)}(z_1,\ldots,z_n) \Big|_{g_s^0} = \frac{1}{4\pi} \int_\gamma \frac{\mathrm{d}z}{z} \left( \prod_{k=1}^n \frac{2z}{z_k^2 - z^2} \right) e^{\frac{\sqrt{2}}{\pi g_s} \sin(2\pi z)}. \tag{B.17}$$

On the other hand, the $\mathcal{O}(g_s)$ term can be easily found by expanding the exponent (B.13), i.e.

$$\hat{W}_n^{(1)}(z_1,\ldots,z_n) \Big|_{g_s} = \frac{\sqrt{2}}{16\pi} \int_\gamma \frac{\mathrm{d}z}{z^2} \left( \prod_{k=1}^n \frac{2z}{z_k^2 - z^2} \right) e^{\frac{\sqrt{2}}{\pi g_s} \sin(2\pi z)}. \tag{B.18}$$

To extract the corresponding one-instanton contribution to the volumes, one just needs to perform an inverse Laplace transforms for each coordinate $z_i$. This can be done by noticing that

$$\int_0^{+\infty} b_i \mathrm{d}b_i \, e^{-b_i z_i} \frac{\sinh(b_i z)}{b_i} = \frac{z}{z_i^2 - z^2}, \tag{B.19}$$

and exchanging the integral over $z$ with the integrals over $z_i$, $i = 1,\ldots,n$. In doing so, we recover the leading order nonperturbative correction to the volumes generating functional:

$$\mathcal{V}_n^{[1]}(b_1,\ldots,b_n) \Big|_{g_s^0} = \frac{2^{\frac{n}{2}}}{4\pi} \int_\gamma \frac{\mathrm{d}z}{z} \left( \prod_{i=1}^n \frac{\sinh(b_i z)}{b_i} \right) e^{\frac{\sqrt{2}}{\pi g_s} \sin(2\pi z)}, \tag{B.20}$$

that coincide with the result (29) of section 2.1. Additionally we also obtain the one-loop perturbative correction $\mathcal{O}(g_s)$:

$$\mathcal{V}_n^{[1]}(b_1,\ldots,b_n) \Big|_{g_s} = \frac{2^{\frac{n+1}{2}}}{16\pi} \int_\gamma \frac{\mathrm{d}z}{z^2} \left( \prod_{i=1}^n \frac{\sinh(b_i z)}{b_i} \right) e^{\frac{\sqrt{2}}{\pi g_s} \sin(2\pi z)}. \tag{B.21}$$

## B.1  The first subleading contribution to the super-volumes

The formulae (B.20) and (B.21) can be used to infer the large genus asymptotic behaviour of the Weil-Petersson volumes. In fact, we can engineer the following ansatz for them:

$$V_{g,n}^{\text{SWP}}(b_1,\ldots,b_n)=\Gamma(2g+n-2)\hat{V}_{g,n}^{(0)}(b_1,\ldots,b_n)+\Gamma(2g+n-3)\ \hat{V}_{g,n}^{(1)}(b_1,\ldots,b_n)+\ldots, \quad (\text{B.22})$$

where the dots stand for subleading corrections diverging like $\Gamma(2g+n-4)$ and so on. For the purposes of this section, in (B.22) we extracted the leading factorial growth for each term in the expansion, but a simple comparison with (11) can reveal the relation $V_{g,n}^{(i)}(b_1,\cdots,b_n) = \Gamma(2g+n-2-i)\hat{V}_{g,n}^{(i)}(b_1,\cdots,b_n)$. We have already determined the leading piece $\hat{V}_{g,n}^{(0)}(b_1,\cdots,b_n)$ in (9) in the main text and now we will carry out a similar analysis for the subleading piece $\hat{V}_{g,n}^{(1)}(b_1,\cdots,b_n)$, that will allow fixing its form completely. In order to do that, we consider the following quantity $\tilde{\mathcal{V}}_n^{[0]}(b_1,\cdots,b_n,S_0)$, given by the difference between the generating function of the volumes and the perturbative sum of their leading piece, i.e.

$$\tilde{\mathcal{V}}_n^{[0]}(b_1,\ldots,b_n,S_0) \equiv \mathcal{V}_n^{[0]}(b_1,\ldots,b_n,S_0)-\sum_{g=0}^{+\infty} g_s^{2g+n-2}\Gamma(2g+n-3)\ \hat{V}_{g,n}^{(0)}(b_1,\ldots,b_n). \quad (\text{B.23})$$

At leading order this quantity explicitly reads:

$$\begin{aligned}\tilde{\mathcal{V}}_n^{[0]}(b_1,\ldots,b_n,S_0) &= \sum_{g=0}^{+\infty} g_s^{2g+n-2}\Gamma(2g+n-3)\ \hat{V}_{g,n}^{(1)}(b_1,\ldots,b_n)\\ &= g_s\int_0^{+\infty}\mathrm{d}t\ \mathcal{B}[\tilde{\mathcal{V}}_n^{[0]}](t)\ e^{-\frac{t}{g_s}},\end{aligned} \quad (\text{B.24})$$

where, in the second step, we have formally integrated term by term the Borel transform $\mathcal{B}[\tilde{\mathcal{V}}_n^{[0]}](t)$ which is defined as:

$$\mathcal{B}[\tilde{\mathcal{V}}_n^{[0]}](t) = \sum_{g=0}^{+\infty} t^{2g+n-4}\ \hat{V}_{g,n}^{(1)}(b_1,\ldots,b_n). \quad (\text{B.25})$$

As outlined in the main text, by knowing the Borel transform $\mathcal{B}[\tilde{\mathcal{V}}_n^{[0]}](t)$, it is then possible to extract the general coefficient $\hat{V}_{g,n}^{(1)}(b_1,\cdots,b_n)$ by means of the Cauchy theorem:

$$\hat{V}_{g,n}^{(1)}(b_1,\ldots,b_n) = \frac{1}{2\pi i}\oint \frac{\mathcal{B}[\tilde{\mathcal{V}}_n^{[0]}](t)}{t^{2g+n-3}}\mathrm{d}t. \quad (\text{B.26})$$

In order to obtain an expression for $\mathcal{B}[\tilde{\mathcal{V}}_n^{[0]}](t)$ we observe that we can bring the one-loop correction (B.21) to the form (B.24) by performing the change of variables $t(z)=-\frac{\sqrt{2}}{\pi}\sin(2\pi z)$. Then the idea is to interpret the resulting expression as the contribution to the Borel resummation of $\tilde{\mathcal{V}}_n^{[0]}(b_1,\cdots,b_n,S_0)$ that encodes the information of the first nonperturbative sector complementing (B.23). We already noticed in the main text that some issues arise when trying to interpret this change of variables at the level of the integration contour, however if we consider these steps as formal, we can immediately read off from (B.21) the Borel transform as $\mathcal{B}[\tilde{\mathcal{V}}_n^{[0]}](t)=f(z(t))\frac{dz}{dt}$ with

$$f(z)=\frac{2^{\frac{n+1}{2}}}{16\pi}\frac{1}{z^2}\left(\prod_{i=1}^{n}\frac{\sinh(b_iz)}{b_i}\right),\quad\text{and}\quad t(z)=-\frac{\sqrt{2}}{\pi}\sin(2\pi z). \quad (\text{B.27})$$

Plugging the these result into (B.25) we get:

$$
\begin{aligned}
\hat{V}_{g,n}^{(1)}(b_1,\ldots,b_n) &= \frac{1}{2\pi i}\oint \frac{1}{t^{2g+n-3}}f(z(t))\frac{\mathrm{d}z}{\mathrm{d}t}\mathrm{d}t \\
&= \frac{1}{2\pi i}\oint \frac{1}{(t(z))^{2g+n-3}}f(z)\mathrm{d}z\,.
\end{aligned}
\tag{B.28}
$$

In the last step, we changed variables from $t$ to $z$. Keeping track of all prefactors, we finally obtain

$$
\hat{V}_{g,n}^{(1)}(b_1,\ldots,b_n) = (-1)^{n+1}\frac{\pi^{2g+n-5}}{2^{g+1-n}i}\oint \frac{1}{4z^2}\frac{1}{(\sin(2\pi z))^{2g+n-3}}\left(\prod_{i=1}^{n}\frac{\sinh(b_i z)}{b_i}\right)\mathrm{d}z\,,
\tag{B.29}
$$

where we have also included a factor of $2^{-n}$ to account for the different normalization between gravity and matrix model observables. We finally remark that an analogous procedure can be done for the leading piece $\hat{V}_{g,n}^{(0)}(b_1,\cdots,b_n)$ by exploiting the comparison with the leading order one-instanton contribution (B.20). This allows us to obtain all correct prefactors that, in the main text, we fixed in a more empirical way to get (9).

## C  Some comments on the numerical implementation of Mirzakhani's recursions

In this appendix, we highlight the crucial ingredients that are needed to efficiently implement in a computer Mirzakhani's recursion for the super-volumes whose form we now display:

$$
\begin{aligned}
bV_g(b,B) = &-\frac{1}{2}\int_0^\infty b'\mathrm{d}b'b''\mathrm{d}b''D(b'+b'',b)V_{g-1}(b',b'',B) \\
&-\frac{1}{2}\int_0^\infty b'\mathrm{d}b'b''\mathrm{d}b''D(b'+b'',b)\sum_{h=0}^{g}\sum_{B_1\subseteq B}V_{h_1}(b',B_1)V_{h_2}(b'',B_2) \\
&-\sum_{k=1}^{|B|}\int_0^\infty b'\mathrm{d}b'[D(b'+b_k,b)+D(b'-b_k,b)]V_g(b',B\backslash b_k)\,,
\end{aligned}
\tag{C.1}
$$

where we denoted with $B$ the set $\{b_1,\ldots,b_n\}$ and with $D(x,y)$ the kernel:

$$
D(x,y) = \frac{1}{8\pi}\left[\frac{1}{\cosh\left(\frac{x-y}{4}\right)} - \frac{1}{\cosh\left(\frac{x+y}{4}\right)}\right]\,.
\tag{C.2}
$$

Similar to the case of the standard Weil-Petersson volumes, the above recursion relations provide a tool to compute each super-volume recursively. As a matter of fact, by utilizing these relations, it is possible to determine every super-volume starting from the initial condition $V_1(b_1) = -\frac{1}{8}$.

Recalling the polynomial structure of the super-volumes, it is clear that the building blocks for an efficient implementation of the recursions in a computer are the following integral formulas:

$$
\int_0^\infty \mathrm{d}x\, x^{2k+1}D(x,t) = F_k(t)\,,
\tag{C.3}
$$

and:

$$
\int_0^\infty \mathrm{d}x\int_0^\infty \mathrm{d}y\, x^{2k+1}y^{2l+1}D(x+y,t) = \frac{(2k+1)!(2l+1)!}{(2k+2l+3)!}F_{k+l+1}(t)\,,
\tag{C.4}
$$

that mimics the integration of a generic monomial pertaining to a super-volume. Here we denoted with $F_k(t)$ the following expression:

$$F_k(t) = \frac{1}{2} \sum_{i=0}^{k} (-1)^{k-i} \binom{2k+1}{2i+1} (4\pi^2)^{k-i} \mathcal{E}_{2k-2i} \, t^{2i+1} \,, \tag{C.5}$$

where $\mathcal{E}_n$ are the usual Euler numbers.

These integrals appeared for the first time in [36] and there it is also explained how to perform them. For the sake of completeness, we report the detailed steps for their computation as well. Starting from (C.3):

$$\begin{aligned}
F_k(t) &= \frac{1}{8\pi} \int_0^\infty x^{2k+1} \left( \frac{1}{\cosh\left(\frac{x-y}{4}\right)} - \frac{1}{\cosh\left(\frac{x+y}{4}\right)} \right) \mathrm{d}x \\
&= \frac{1}{8\pi} \int_{-t}^\infty \frac{(x+t)^{2k+1}}{\cosh\left(\frac{x}{4}\right)} \mathrm{d}x - \frac{1}{8\pi} \int_t^\infty \frac{(x-t)^{2k+1}}{\cosh\left(\frac{x}{4}\right)} \mathrm{d}x \\
&= \frac{1}{8\pi} \int_0^\infty \frac{(x+t)^{2k+1} - (x-t)^{2k+1}}{\cosh\left(\frac{x}{4}\right)} \mathrm{d}x \\
&= \frac{1}{4\pi} \sum_{i=0}^k \binom{2k+1}{2i+1} t^{2i+1} \int_0^\infty \frac{x^{2k-2i}}{\cosh\left(\frac{x}{4}\right)} \mathrm{d}x \\
&= \frac{1}{2} \sum_{i=0}^k \binom{2k+1}{2i+1} a_{k-i} t^{2i+1} \,,
\end{aligned} \tag{C.6}$$

where $a_n$ is defined by $\frac{1}{\cos(2\pi x)} = \sum_{n=0}^\infty a_n \frac{x^{2n}}{(2n)!}$ and can be expressed in terms of the Euler numbers:

$$a_n = (-1)^n (4\pi^2)^n \mathcal{E}_n \,. \tag{C.7}$$

The integral in (C.4) can be performed by the same methodology of (C.3) as soon as one performs the change of integration variables $x = u + v$, $y = u - v$. Following an analogous procedure to the one that we showed above, one quickly recovers the expression on the right-hand side of (C.4).

Using the previous results, we slightly modified the Mathematica notebook attached to [39] to efficiently compute the super-volumes whose explicit expression was needed for the numerical check of section 2.1.

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
