# Peer review of "Asymptotics of Weil-Petersson volumes and two-dimensional quantum gravities"

_SciPost Physics, doi:SciPost Phys. 17, 156 (2024)_

## Round 1 · Referee Report · Anonymous (Referee 1) · 2024-7-14

Report

In this work, the authors propose a new formula for the large genus asymptotics of the super-Weil-Petersson volumes generalizing the conjecture of Witten and Stanford. The paper provides a formal argument and various justifications for this conjecture. Numerical evidence and applications are also provided.

The paper is clear and lucid. The introduction does a good job of motivating the problem and providing sufficient background. The main arguments and derivations are presented clearly. The appendices provide relevant details. Although the authors state that the derivations are formal in various places, there are some places where this can be stated more clearly.

I recommend that the paper be published after making some very minor changes.

Requested changes

(1) An important piece of evidence justifying the proposal is the recurrence relation in eq. (2.6). It would be helpful if the authors could include a few sentences explaining the derivation of this relation for completeness.

(2) Below eq. (2.8), the authors state that “In analogy to (2.5), this expression results in a polynomial of degree g − 1 in b_i^2, and its explicit form is derived in Appendix B through an examination of the contributions from one-eigenvalue instantons to the n-point correlators in SJT gravity.”

This appears to gloss over the fact that the derivation in Appendix B is formal. I suggest that the wording be changed to indicate this.

(3) Below eq. (2.29), the authors state that “For those concerned about the perceived lack of systematicity in the final part of this section, we will address this issue in Appendix B. There, we refine our analysis and introduce a more systematic approach for computing instantonic contributions in a matrix integral [15]. Importantly, by following this method, we successfully replicate the result presented in (2.5) with all the accurate prefactors.”

This seems to gloss over the fact that the derivation in Appendix B is formal. I suggest that the wording be changed to indicate this.

(4) The derivation in Appendix B is formal for two main reasons - the integration contour γ is not known explicitly and the change of variables from t to z is not justified. Although the authors say this in the details of the computations, it would be helpful if this is reiterated at the beginning of the appendix.

Recommendation

Ask for minor revision

  • validity: high
  • significance: good
  • originality: good
  • clarity: high
  • formatting: excellent
  • grammar: excellent

Author:  Jacopo Papalini  on 2024-10-30  [id 4916]

(in reply to Report 1 on 2024-07-14)

We thank the Referee for his valuable suggestions and remarks. Below, we will address the issued raised by the the Referee.

  1. We added few lines below (2.6) to sketch the ingredients for its proof.

  2. Below (2.8) we stressed that the derivation given in appendix is only formal

  3. Below (2.30) we stressed again that the derivation given in appendix is only formal

  4. At the beginning of Appendix $B$ we have reiterated the fact that subsequent derivation is only formal

---

## Round 1 · Referee Report · Anonymous (Referee 2) · 2024-8-28

Report

The main result of this paper is equation (2.5), a prediction for the leading large g behavior of (super) Weil-Petersson polynomials. The formula is elegant and the fact that it retains the polynomial structure may be useful for physical applications.

The main technical difficulty is to argue for equation (2.7). There is a derivation in section 2.2 which to me seems sufficiently rigorous for any hep-th paper. An allegedly more rigorous derivation is presented in appendix B.

A negative point is that many of the technical steps required in the derivation are not explained. It is next to impossible to follow / check the derivation without consulting the references. For instance, equation (2.24), which is very important, comes out of the blue. Similarly, equation (B.1) and (B.7) come out of the blue. This is annoying, as the equations that I mention are in teh respective sections the main technical equations that are used to derive (2.7). This leaves the reader with very little intuition about the physical meaning of the result.

A second comment is that rather little effort is being made to physically motivate why one would be interested in these large genus assymptotics (except for brief comments in the introduction and in the end of the concluding remarks). As such, one could wonder if this paper is a maths paper or an hep-th paper.

A positive is that the general build-up of the paper is nice. The combination of two rather rigorous calculations with numerical evidence convinces the reader of the validity of the main equation (2.5), without having to get lost in details such as how to precisely derive the integration contour. The validity is obvious from the combination of all the sections.

The improvement of (3.14) as compared to (3.13) is impressive.

The paper is in general enjoyable to read.

The paper is certainly worthy of publication, as the main equation (2.5) looks interesting and is correct. Before that, however, I would appreciate it if the authors could briefly think about my questions below and make a few changes. The questions are the following

  1. Would (2.5) workfor generic dilaton gravity models with the relevant y(z) inserted? If not, which feature of the spectral curve would present a limitation? Were these equations known for ordinary JT gravity (n=1 can be found in SSS) or not? If not, it would be worth emphasizing this.

  2. I am lacking a gravitational interpretation of these calculations. Is there a precise relation with ZZ branes, perhaps even just for the saddle-points of the z integrals?

  3. Regarding the comment at the end of the discussion, is there a sense in which the large g behavior is sufficient to reproduce certain salient features of the spectral form factor at late times? This does not seem obvious, but it does seem plausible given the universality of equation (2.5) (assuming the answer to question 1 is positive).

Requested changes

  1. The paper is not self-contained, not even approximately. The crucial equations (2.24), (B.1) and (B.7) should be motivated in a more-or-less self consistent way in the paper, such that the reader can judge the validity of the calculations without having to know all of the litterature. It would be useful (though in the context of the general style of the paper not crucial) to have gravitational intuition for these equations too. This qualifies as a major modification, though not one that should require too much effort from the side of the authors.

Recommendation

Ask for major revision

  • validity: high
  • significance: high
  • originality: good
  • clarity: ok
  • formatting: perfect
  • grammar: excellent

Author:  Jacopo Papalini  on 2024-10-30  [id 4915]

(in reply to Report 2 on 2024-08-28)

The response to the report is provided in the attached file, where we've included formulas and references for clarity.

Attachment:

Reply_to_the_Report.pdf

---

## Editorial Decision

published